# Assessment of emission scenarios for 2030 and impacts of black carbon emission reduction measures on air quality and radiative forcing in Southeast Asia

Didin Agustian Permadi[1], Nguyen Thi Kim Oanh[1*] and Robert Vautard[2]

[1] Environmental Engineering and Management; School of Environment, Resources and Development; Asian Institute of Technology; Klong Luang, Pathumthani 12120, Thailand.

[2] Laboratoire des Sciences du Climate de l'Environment (LSCE), Institut Pierre Simon Laplace (IPSL), Gif Sur Yvette, France.

Correspondence to: Nguyen Thi Kim Oanh (kimoanh@ait.ac.th)

**Abstract.** Our accompanying paper (Permadi et al., 2017a) focuses on the preparation of emission input data and evaluation of WRF/CHIMERE performance in 2007, this paper follows with detailing the impact assessment of the future (2030) black carbon (BC) emission reduction measures for Southeast Asia (SEA) countries on air quality, health and BC direct radiative forcing (DRF). The business as usual (BAU2030) projected emissions from the base year of 2007 (BY2007) assuming "no intervention" with the linear projection of the emissions based on the past activity data for Indonesia and Thailand, and the sectoral GDP growth for other countries. The RED2030 featured measures to cut down emission in major four source sectors in Indonesia and Thailand (road transport, residential cooking, industry, biomass open burning) while for other countries the representative concentration pathway 8.5 (RCP8.5) emissions were assumed. WRF/CHIMERE simulated levels of aerosol species under BAU2030 and RED2030 for the modeling domain using the base year meteorology and 2030 boundary conditions from LMDZ/INCA. The extended aerosol optical depth module (AODEM) calculated the total columnar AOD and BC AOD for all scenarios with an assumption on the internal mixing state. Under RED2030, the health benefits were analyzed in term of the avoided number of premature deaths associated with ambient $PM_{2.5}$ reduction along with BC DRF reduction. Under BAU2030, the average number of the premature deaths per 100,000 population in the SEA domain would increase by 30 from BY2007 while under RED2030 the premature deaths would be cut-down (avoided) by 59 from the RED2030. In 2007, the maximum annual average BC DRF in the SEA countries was 0.98 W m$^{-2}$ which would increase to 2.0 W m$^{-2}$ under BAU2030 and 1.4 W m$^{-2}$ under RED2030. Substantial impacts on human health and BC DRF reduction in SEA could be resulted from the emission measures incorporated in RED2030. Future works should consider other impacts such as for the agricultural crop production as well as the cost benefit analysis of the measures implementation to provide relevant information for policy making.

## 1. Introduction

The interaction between aerosol (fine particles suspended in the atmosphere) and climate has gained an increasing attention from the scientific community, especially to assess various emission control measures for near-term climate change mitigation. Being black in color, in the atmosphere the black carbon (BC) particles strongly absorb solar radiation, hence exerting a positive direct radiative forcing. BC is known as a short-lived climate forcing

pollutant (SLCP) because of its short atmospheric lifetime of a few days to weeks as compared to the long-lived carbon dioxide, for example. BC interacts with the cloud formation processes and once deposits on snow it reduces the surface albedo and consequently affects the Earth's radiation energy balance (Myhre et al., 2001). Several global modeling studies estimated the present-day BC radiative forcing of $+0.2 - +1.1$ W $m^{-2}$ hence BC has been

recognized as the second most important global warming agent after $CO_2$ (Bond et al., 2013; Ramanathan and Carmichael, 2008). Globally, measures aim to reduce emissions of BC (and co-emitting pollutants) have been shown to reduce the number of premature deaths and slow down the temperature increase rate in the near future, i.e. bring in co-benefits, and more to be gained in Asia, where current emissions are high (UNEP-WMO, 2011; Shindell et al., 2012).

Southeast Asia (SEA) has high emissions from anthropogenic sources that contribute significantly to the Asia and global emissions (Permadi et al., 2017a). Overall, Asian emissions have been reported to increase rapidly over last decades (Streets et al., 2003; Zhang et al., 2009; Ohara et al., 2007; EJ/JRC-PBL, 2010). High levels of air pollution, especially the fine particles or $PM_{2.5}$ (particles with aerodynamic diameter $\leq2.5$ um) that caused severe health effects (WHO, 2012) have been measured in Asian cities (Kim Oanh et al., 2006; Hopke et al., 2008).

Global studies reported that efforts to reduce SLCP emissions (BC and ozone precursors) would help to reduce global warming immediately (UNEP and WMO, 2011; Shindell et al., 2012) which should complement those addressing the long-lived greenhouse gases (GHGs) that require longer time to realize. Developing Asia is also vulnerable to climate change hence reducing emissions of the SLCPs would bring in co-benefits in terms of avoiding excessive premature deaths, reducing the crop yield loss, as well as in slowing down the temperature

increase rate (UNEP-WMO, 2011).

The co-benefit approach is being more and more recognized as an important concept to simultaneously address the problems of air pollution and climate change. However, so far it has not been adequately considered in the policy making in many developing countries. A number of studies reported applications of the modeling tool to investigate impacts of emission reduction measures on the premature mortality and radiative forcing in East and

South Asia (Saikawa et al., 2009; Aunan et al., 2006; Akimoto et al., 2015). There are no such detail studies conducted for the SEA region where the local/domestic and transboundary problems, e.g. regional haze, are important (Heil and Goldammer, 2001).

In this paper we focus on the simulations of aerosol concentrations in SEA under two emission scenarios, projected for 2030, using the chemistry transport model (CTM) of CHIMERE (Vautard et al., 2001; Bessagnet et al., 2004).

Our accompanying paper (Permadi et al., 2017a) has detailed the SEA emissions and the evaluation of WRF/CHIMERE performance for $PM_{10}$, $PM_{2.5}$ and BC in SEA in the base year of 2007 (BY2007). The emission reduction scenario (RED2030) considered reduction measures to be implemented in four major anthropogenic source sectors (road transportation, residential cooking, industry and biomass open burning) in two large emitting SEA countries of Thailand and Indonesia. The changes in the BC direct radiative forcing (DRF) and in the number

of avoided premature deaths between BY2007 and RED2030 were compared to those between BY2007 and the business as usual scenario (BAU2030) to highlight potential impacts. The results of this study would provide information to policy makers on the efficacy of different emission reduction measures and associated benefits for improving air quality, reducing health effects, and mitigating BC DRF in SEA. To our best knowledge, this is the

first study addressing air quality and BC DRF impacts for the SEA region hence the results would contribute scientific evidences to promote the co-control approach that is currently not incorporated in the policy in any country in the region.

## 2. Methodology

### 2.1 Model simulation

Details of the models used and their configurations have been presented in the accompanying paper (Permadi et al., 2017a). The emission inputs were prepared based on two developed emission scenarios (BAU2030 and RED2030). The meteorological fields produced by WRF for 2007 were used for the simulation of scenarios in 2030. The base year of 2007 was selected because it was not affected by the strong El Niño and La Niña events (http://www.cpc.noaa.gov/products/analysis_monitoring/ensostuff/ensoyears.shtml). We also took the advantage of the satisfactorily evaluated WRF performance for 2007 to examine the effects of the emission reductions alone under the scenarios. However, to account for the changes in the boundary conditions in the simulations of future emission scenarios, we used the chemistry boundary conditions obtained from the global chemistry–aerosol–climate model that couple online the LMDz (Laboratoire de Météorologie Dynamique, version 4) general circulation model (Hourdin et al., 2006) and the INCA (INteraction with Chemistry and Aerosols, version 3) model (Hauglustaine et al., 2014) using the global RCP8.5 emissions (Hauglustaine, 2013). Specifically, the concentrations of 27 pollutants, including aerosol and trace gases, as the SEA domain boundary conditions (monthly average), were extracted from the global LMDZ/INCA simulation for the base year 2007 (BY2007) and 2030 (BAU2030 & RED2030) for 19 hybrid vertical pressure levels. The horizontal resolution is 1.9º in latitude and 3.75º in longitude of LMDZ/INCA output. Boundary conditions from the LMDz/INCA were processed using the available routine in CHIMERE to read the monthly concentrations and get the information of the lateral (lat_con) and top boundary (top_con) concentrations. We used the ratio of simulated levels between 2030 and 2007 for each species to estimate the boundary conditions in 2030 and modified the model inputs. Accordingly, using the output levels of LMDZ/INCA for BC and OC for the year of 2007 and 2030 we estimated the ratios of each BC and OC species between these 2 years (2030/2007 ratios) and presented in Figure S1, SI. Note that the biogenic emissions in 2030 were assumed to be the same as in 2007.

In the discussion of modeling results throughout the paper, two domains are defined as follows: 1) the modeling domain: covering the SEA countries and the southern part of China, and 2) the SEA domain: covering only the nine (9) SEA countries in the domain. Note that, base year emission of East Timor was not provided by the Center for Global and Regional Environmental Research (CGRER) hence it was not included in the SEA domain.

### 2.2 Emission scenarios

BAU2030: This is a reference future scenario, named "Business as Usual", representing the emissions in 2030 (BAU2030). BAU was developed based on the assumption that the emissions would grow following the current trend of the activity data. For Indonesia and Thailand, two large emitting countries in SEA, the historical activity data available prior to 2007 was collected and the trends were examined over the period of the available data of

4-19 years (varying with emission source sector). Further, a simple regression analysis was conducted to analyze the linear relationship between the emissions and a selected proxy which was used to project the activity data to 2030 for the respective country. In principle, it is desirable to have long historical data periods for the emission projection (for Thailand and Indonesia). However, in 1997-1998 the region went through a drastic economic recession hence inclusion of the data before 1999 may induce a large bias. Therefore, even when a longer data period available for a sector we still relied on the trend obtained for the period of 1999-2007. Note that only a short period of historical data (4 years) available for the petroleum refinery and mining production sectors in Thailand hence it may contribute more uncertainty to the BAU scenario emission results. Nevertheless, these sectors did not contribute significantly to the total emissions. The same EFs of 2007 were used for the emission estimation from the sources in 2030 and this may contribute a certain uncertainty to the projected emissions. In general, we should expect the EFs to be reduced in the future with the progressive technology intrusion. Nevertheless, if the vehicle age limit is not strictly enforced for then more aged engines would have higher EFs in the future. For other countries/territories in the modeling domain, due to the limitation of historical activity data, the BAU emissions were projected using their GDP growth trends over the period 2000-2007. The BY2007 emission data for those countries was taken from the international databases that were provided at a different resolution (i.e. 0.5º x 0.5º). Therefore, the spatial analysis of "grid conversion" in Geographical Information System (GIS) was used to convert the emission data in to the same grid resolution 30 x 30 km$^2$ for the whole domain. The land mask map that was generated from WRF model with a resolution of 30 x 30 km$^2$ was used to regrid the emissions in the border between land and sea (assigning zero emissions to the sea area). The same spatial and temporal emission distribution patterns of 2007 were used in the BAU2030 emissions for the whole modeling domain.

RED2030: This scenario considers the PM emission cut for major emission sources in 2030. For Indonesia and Thailand, RED2030 emissions were calculated using the available official policy documents that mainstreamed both air quality and climate change mitigations in four major emission sectors (road transport, residential, industry, and biomass open burning) as detailed in Table S1. These involved cleaner transportation fleets with at least Euro2 in Indonesia and Euro4 in Thailand for personal cars, while NG gas should be used in all public buses in Indonesia, and all public buses and taxis in Thailand. The measures considered in residential cooking for Indonesia included the national program of conversion of kerosene to LPG for cooking (Permadi et al., 2017b) and introduction of cleaner biomass fuel (for gasifier cookstove) to replace the traditional fuel wood cookstoves. In Thailand, cleaner fuel, such as LPG was introduced to replace fuel wood and charcoal, as well as the implementation of rural electrification to enable application of electric cookstoves. In the industrial sector in Indonesia, fuel switching and process modernization were assumed for key industries such as cement, iron steel, pulp and paper and textile (ICCSR, 2010). In Thailand, measures focused on energy saving program and maximum feasible reduction in the key industries, such as cement and iron steel industry, were considered following the policy action proposed by Chotichanatawewong and Thongplew (2012). For biomass open burning, Indonesia focused on the reduction of forest burning area target and zero burning of solid waste following the National Strategic Plan document (MoF, 2010). A clear target was set in Thailand to follow the National Masterplan of Open Burning Control (PCD, 2010) which mandated that forest area would not be burned over than 48,000 ha yr$^{-1}$ in 2030 and zero burning for crop residue implemented throughout the country.

For other countries in the modeling domain, due to the lack of relevant information, this study assumed that their emissions followed the RCP8.5 pathways (taken from http://tntcat.iiasa.ac.at:8787/RcpDb/) for the Asian region. Accordingly, under RCP8.5, in 2030 the SEA emissions of $CH_4$ and $NH_3$ would increase by 1.7 and 1.3 times, respectively, as compared to 2007, while NMVOC remained nearly the same during the period. For other pollutants ($SO_2$, CO, NOx, BC, and OC), some emission reductions were expected, i.e. 2030 emissions were 0.6-0.9 of the respective 2007 emissions. Note that the same temporal and spatial emission distribution patterns of BY2007 were used also in RED2030 emissions.

## 2.3    Assessment of impacts on air quality and BC direct radiative forcing

The potential impacts of the emission reduction scenarios on improvement of air quality (hence health benefits) and mitigation of climate forcing were assessed and quantified. Impacts of emission reduction measures under RED2030 as compared to the reference scenario (BAU2030) were analyzed based on the number of premature deaths and BC DRF in SEA.

### 2.3.1 Premature death

The number of avoidable premature deaths resulting from the emission reduction measures was quantified based on the changes in the ambient $PM_{2.5}$ levels and was calculated using eq. 1 (Wang and Mauzeral, 2006; Saikawa et al. 2009).

$$cases = I_{ref} \cdot POP \cdot CR \cdot C \tag{1}$$

Where,

$\Delta_{cases}$ :        Additional change in number of cases of mortality per year due to change in the annual ambient $PM_{2.5}$ concentration.

$I_{ref}$ :        baseline mortality rate (%)

$POP$ :        number of exposed population (person)

$CR$ :        concentration response coefficient (CR) for mortality rate (unit is % change in mortality and morbidity as a result of a 1 µg m$^{-3}$ change in annual average $PM_{2.5}$ concentration)

$\Delta C$ :        change in annual ambient $PM_{2.5}$ concentration under a given different emission scenario.

We considered only the premature deaths because of the less comprehensive information available on the association between the morbidity and air pollution exposure as compared to that of the mortality (Saikawa et al., 2009). The $PM_{2.5}$ annual concentration was used rather than BC alone following the recommendations of the WHO Task Force (WHO, 2012) that $PM_{2.5}$ should continue to be used as the primary metric in quantifying the human exposure to PM and resulting health effects. Concentration response (CR) data was obtained from Smith et al. (2009) who indicated that every increase in $PM_{2.5}$ by 1 µg m$^{-3}$ is approximately associated with an 1.006% increase in the risk of all cause adult mortality for individual of 20 years and older. The $I_{ref}$ data representing the adult mortality rate for all causes and both sexes (per 1,000 of population) for every nation included in the study domain was obtained from the United Nations database (http://data.un.org/Data.aspx?d=WHO&f=MEASURE_CODE%3AWHOSIS_000004) for the base year of 2007.

The gridded population (*POP*) data of 0.5° resolution was obtained from the Potsdam Institute for Climate Impact Research (PICIR) (http://clima-dods.ictp.it/d10/fcolon_g/) for both 2007 and 2030, and the data were further gridded to a finer resolution of 0.25° of the modeling domain. The proportions of adult population (age ≥20 years) for the countries were taken from the World Bank database (http://data.worldbank.org/indicator/SP.POP.0014.TO.ZS). This study assumed that the baseline mortality and age structure were distributed following the population density and were not changed between 2007 and 2030.

**2.3.2 Black carbon aerosol optical depth**

The BC aerosol optical depth (AOD) values were calculated using AODEM with the 3D aerosol concentration field generated by WRF/CHIMERE for BAU2030 and RED2030. The application detail of AODEM software and parameterization has been presented in our accompanying paper (Permadi et al., 2017a).

**2.3.3 BC direct radiative forcing**

To estimate the change in radiative forcing (*ΔF*), equation (eq. 2) was used following the approach presented in Chylek and Wong (1995) and Kim et al. (2012).

$$F = \frac{1}{4}F_0 T^2 (1 - A_c)\{4R\tau_{ab} - 2((1-R)^2 \beta \tau_{sc}\} \tag{2}$$

Where, $F_0$ is the solar irradiance constant (1,370 W m$^{-2}$) and $T$ is the atmospheric transmission coefficient (0.79). The values of $A_c$ (the total cloud fraction) and $R$ (surface albedo) were taken from the WRF simulation results, respectively. The expression of (*1-Ac*) represents the clear sky conditions, i.e. we consider only clear sky BC DRF. The expression of *(1-R)$^2$* in the equation accounts for the multiple reflection of the aerosol layer. The surface $\tau_{ab}$ is the absorption (BC AOD) and $\tau_{sc}$ is the scattering optical depths calculated by AODEM based on the BC burden, $\beta$ is the back-scattering fraction which was assumed to be 0.17 based on the measurements of Schnaiter et al. (2003). Note that, in the current vertical model set-up the top of the model domain was 500 hPa hence the simulation results may not be able to capture the long-range transport (LRT) that took place above the domain top in the free tropospheric layer and subsequently underestimate BC AOD and BC DRF. However, above the domain top (>5 km) the biomass open burning emission of importance in the region may not have significant effects on the aerosol extinction coefficient as shown by the Cloud-Aerosol Lidar with Orthogonal Polarization (CALIOP) observations (Campbell et al., 2013). Therefore, the uncertainty caused by the vertical model set-up on the simulation results may not be significant considering the biomass open burning emissions in particular.

**3. Results and Discussion**

**3.1 Emission scenarios**

A summary of the annual emissions of key species in the base year of 2007 and in 2030 under two emission scenarios (BAU2030 and RED2030) from the modeling domain is given in Table 1. The annual emission changes under the scenarios are indicated by the respective ratios for each species. This section discussed in depth the

emissions for 2 countries, Indonesia and Thailand, for which the specific activity data trends were considered for the emission scenarios development. Table S2, SI showed the increases in all activity data of the key anthropogenic sources in the countries except for forest fire emissions, which were actually expected to reduce due to a lower GDP from the forestry sector and less forest biomass/area available in the future. The most significant increase was for the number of registered vehicles in Indonesia which in 2030 would be 3.1 times above that in 2007. In Thailand, the petroleum refinery product was also projected to increase by 3.3 times from the 2007 level, followed by the number of vehicles, by 2.2 times. The sectoral emissions of key species from these 2 countries under different scenarios are provided in Figure S2, SI.

BAU2030 emissions: For Indonesia, under BAU2030, the emissions of every species would increase with a ratio of 1.2-1.95 between BAU2030 and BY2007 (Table 1). The highest increase (1.95 times) would be expected for $SO_2$ and this is mainly reflecting the increase in coal usage for the power generation and industry. Note that, the use of the same $SO_2$ EFs for BY2007 and BAU2030 emission calculation, as well as the assumption on no further emission control devices applied in the power and industry may be the cause of this high increase rate for $SO_2$. This may not represent the actual trend because the regulation on emission control would be more stringent in the future hence less $SO_2$ increase should be expected. The NOx emission increases by 1.6 times was mainly due to the increasing trends of the oil and gas industry, and traffic activities. Because of the dominant contributions from the residential combustion to their emissions, the population growth in the country during the period of 2007 – 2030 would increase emissions of CO, NMVOC, and PM ($PM_{10}$, $PM_{2.5}$, BC, OC) by 1.2, 1.5 and 1.35 – 1.39 times, respectively. $NH_3$ emission was projected to increase by 1.6 times due to the livestock growth in the country. Among GHGs, $CO_2$ would increase by 1.5 times while $CH_4$ would increase by 1.4 times from the base year emissions.

For Thailand, the BAU2030/BY2007 ratios showing the increases of all species in Table 1, highest for $SO_2$ (1.95 times) and lowest for CO (1.01 times). The increases in $SO_2$ and NOx emissions reflected the increasing rates of the coal consumption in particular and that is related to the ratio of projected energy consumption 2030/2007 of 2.1 for industry (EPPO, 2008), as seen in Table S2, SI. The intensification of agricultural production would increase the crop residue OB that caused the increased emissions of PM species of $PM_{10}$, $PM_{2.5}$, BC, OC by 1.42-1.66 varying with species. Forest fire emission was projected to reduce with estimated 2030/2007 ratios of 0.7 (Indonesia) and 0.9 (Thailand) which were based on the trend of GDP in forestry (Indonesia) and the trend of forest area (Thailand). The emission increases for $NH_3$ (1.55) and NMVOC (1.54) were mainly due to the livestock and industrial growth rates, respectively. GHGs emissions were also projected to increase by 1.10 ($CO_2$), 1.61 ($CH_4$) and 1.81 ($N_2O$). The substantial increases of activity data levels under BAU2030 for both countries have brought about the increase in emissions for all species in both countries from the BY2007 levels.

The emission projection under BAU2030 for other countries in the modeling domain was made based on the GDP growth rate that was, for example, as high as 10.5-11.5% for Cambodia, China and Myanmar (WB, 2008) over the period of 2000-2007. The BAU2030 emissions from other countries (than Indonesia and Thailand) in the modeling domain were assumed to grow following their population weighted average GDP growth rates, averaged at 2.2 times. As compared to BY2007, the regional BC and $CO_2$ emissions under BAU2030 in 2030 increased by

about 1.89 and 1.41 times, respectively, and were well above the increase rate specified by the Intergovernmental Panel on Climate Change (IPCC) RCP6.0 for BC and $CO_2$ of 1.03 and 1.23 times for SEA.

RED2030 emissions: For Indonesia, the measures to be implemented in the RED2030 scenario would result in lower emissions as compared to BAU2030, with the ratios RED2030/BAU2030 being 0.3-0.92. The most significant emission reductions would be realized for aerosol species ($PM_{10}$, $PM_{2.5}$, BC, and OC) with the RED2030/BAU2030 of 0.30-0.56 followed by CO (0.62) as seen in Fig. S2, SI. The significant reductions for these species were expected because the included measures (Table S1) in all sectors aimed at reducing PM (BC) emissions which also would reduce CO, an accompanying gas emitted from these combustion sources. Among GHGs, the highest reduction was for $CO_2$ with RED2030/BAU2030 ratio of 0.7 while lower reductions were shown for $CH_4$ (0.96) and $N_2O$ (0.87). As compared to the base year, RED2030 would also feature more emission reductions for major aerosol species with the corresponding ratios of RED2030/BY2007 of 0.4-0.75 for PM species and 0.74 for CO. The RED2030 emissions were lower than the base year of 2007 for most species except for $SO_2$, $NH_3$, NMVOC and $NO_x$ which were increased by 1.2 – 1.8 times.

For Thailand, RED2030 also featured the lower emissions than BAU2030, as expected. The reductions were about in the same ranges as for Indonesia. The emission ratios of RED2030/BAU2030 for different species were 0.37-0.93. More significant reductions were obtained for PM species, i.e. ratios of 0.37-0.55, followed by CO, a ratio of 0.73. Among the GHGs, the highest reduction would be for $CH_4$ with the ratio of 0.81 followed by $CO_2$ (0.84) and $N_2O$ (0.95). Similarly to the case of Indonesia, the emissions under RED2030 as compared to BY2007 were lower for PM species and CO, but higher for other species with the RED2030/BY2007 ratios of 1.2 – 1.6 (Table 1).

Sector-wise emissions in both countries showed that under RED2030 there would be a substantial reduction, as compared to BAU2030, resulted from the mitigation measures (Table S1) in the two countries for all sectors. The emission ratios of RED2030/BAU2030 for biomass OB were 0.09-0.6 while for residential sector corresponding ratios were 0.14 – 0.99 for different species (Fig. S2, SI). There observed some increasing trends of $SO_2$ emissions from the residential sector when LPG was used to replace wood as cooking fuel because of its higher sulfur content. As for BC species, the emission ratios of RED2030/BAU2030 for biomass OB and residential sector in Indonesia were 0.48 and 0.49, respectively, and in Thailand were 0.37 and 0.41, respectively. For the industry and transport sectors, the corresponding emission ratios for BC were 0.87 (Indonesia) and 0.6 (Thailand). Under BAU2030, the emissions of BC was rising steadily with a high rate from the BY2007 (Fig. S2, SI).

For other countries in the modeling domain, the IPCC RCP8.5 was used to project the emissions under RED2030 and this was done using the regional (Asia) emission ratios between 2030 and 2007 emissions (using the IPCC emission database) for every species. As discussed above, the RED2030 featured the emission reductions from BY2007 for several species presented in Table 1, i.e. with the RED2030/BY2007 ratios of 0.84-0.88 for PM species, 0.69 for $SO_2$ and 0.98 for CO. The emissions of NOx, $NH_3$ and NMVOC in these countries were however increased by 1.05-1.3 times (Table 1).

Due to the insufficient information on air quality and climate policies of other countries in the modeling domain, this study focused primarily on the emission reduction measures for 2 countries of Indonesia and Thailand.

Nevertheless, as highlighted in our accompanying paper (Permadi et al., 2017a), collectively these two countries had the large contributions in the domain, i.e. sharing 25-66 % of the total emissions from the SEA domain and 17-44% of the total modeling domain (SEA + southern part of China) for most of the species in 2007. This justified the importance of the emission reduction measures of these 2 countries for the SEA region. Note that the

reductions for $NH_3$, $CH_4$ and $N_2O$ were relatively small in both countries because the major sources of these species were not addressed in the four source sectors considered for mitigation in this study. On the opposite, the reductions PM species and CO were relatively more significant for both countries because the mitigation measures addressed the key sources of the species. It is interesting to note that our results for the national BC emission ratios between RED2030 and BY2007 for Indonesia (0.74) and Thailand (0.81) were close to the IPCC RCP2.6 scenario

for the 2 countries of 0.82 and 0.81, respectively. For $CO_2$, the emission ratios were 0.7 and 0.84 which were slightly lower than the IPCC RCP2.6 values of 0.92 and 0.86, respectively. It therefore shows that the RED2030 is very much aligning with the IPCC RCP2.6 for the countries and therefore suggesting that the current master plans in the two countries could lead to achieving the 2.0º target.

**3.2 Impacts assessment of emission reduction measures in 2030**

Our impacts assessment of emission reduction measures in 2030 covered the health impact in term of the avoided number of premature deaths associated with the reduced $PM_{2.5}$ pollution and the reduction in BC DRF. The results of RED2030 were compared to those of BAU2030 to quantify the impacts of the emission reduction measures.

**3.2.1 Air quality and premature deaths**

1) PM and BC

A summary of the maximum $PM_{2.5}$, $PM_{10}$ and BC along with BC DRF and premature deaths under the scenarios is given in Table 2 for the SEA domain only, i.e. excluding the southern China part of the modeling domain. Annual average concentrations of BC and $PM_{2.5}$ for BY2007 and two different scenarios of BAU2030 and RED2030 are presented in Figure 2, which shows similar patterns of BC and $PM_{2.5}$ for all scenarios but their maximum values were much higher under BAU2030 as compared to other scenarios. Over the SEA countries, the

maximum BC and $PM_{2.5}$ were shown over the eastern part of Java Island (Indonesia) in all scenarios. The maximum BC values simulated for BAU2030 of 7.2 µg $m^{-3}$ appeared over the eastern part of Java Island that featured an increase by 1.2 µg $m^{-3}$ from the BY2007. Under RED2030, the maximum of BC in the SEA domain was 4.3 µg $m^{-3}$ which is 2.9 µg $m^{-3}$ lower than that under BAU2030.

For $PM_{2.5}$, under BAU2030 there was an increase of the maximum level over the SEA domain, from 32 µg $m^{-3}$ in BY2007 to 36 µg $m^{-3}$. The emission reduction measures under RED2030 would reduce the SEA domain maximum $PM_{2.5}$ to 21 µg $m^{-3}$ (Table 2). The simulated maximum hourly concentrations of $PM_{2.5}$, $PM_{10}$, and BC under BY2007 were 189, 327 and 39 µg $m^{-3}$, respectively, that increased to 296, 472, and 59 µg $m^{-3}$, respectively, under the BAU2030. Measures implemented under RED2030 would reduce the hourly maximum concentrations of

$PM_{2.5}$, $PM_{10}$, and BC to 146, 247, 32 µg $m^{-3}$ (Table 2). The hourly maximum concentrations of $PM_{2.5}$ and $PM_{10}$ occurred in the Borneo Island during the intensive period of biomass open burning while that for BC occurred over the eastern part of Java Island. A sharp increase in the maximum hourly concentrations under BAU2030 (e.g.

1h $PM_{10}$ reached 472 µg m$^{-3}$) also occurred in the Borneo Island where the emissions from crop residue open burning were assumed to be intensified.

The changes in the total BC and $PM_{2.5}$ emissions in the domain were consistent to the emission changes under different scenarios. As discussed above, BAU2030 featured a total emission increase in the $PM_{2.5}$ emissions (primary particles) in the SEA domain by 1.6 times as compared to BY2007 and in the BC emissions (1.6 times). In 2030, RED2030 would reduce the emissions of $PM_{2.5}$ (primary) by 2.4 times and BC emission by 2.1 times as compared to BAU2030. The ambient $PM_{2.5}$ should compose also of secondary particles hence would be also affected by the changes in the precursors' emissions. The magnitude of changes in $PM_{2.5}$ and BC emissions were not similar to the changes in their simulated ambient concentrations suggesting influences of atmospheric processes of dispersion, removal/wet scavenging as well as the boundary conditions used for 2007 and 2030. Overall, the emission reduction measures implemented in four anthropogenic source sectors under RED2030 could maintain the $PM_{2.5}$ and BC levels in 2030 quite close to or even lower than those in 2007 while under BAU2030 their levels would substantially increase.

2) Number of premature death

The impacts of the two emission scenarios on the number of the premature deaths in 2030, i.e. BAU2030 vs. BY2007, and RED2030 vs. BAU2030, were quantified based on the ambient $PM_{2.5}$ and the results (the difference) are presented in Figure 3. The increase in annual $PM_{2.5}$ levels under BAU2030 compared to BY2007 would bring in the total number of additional premature deaths in the SEA countries of around 201,000 cases (average 30 per 100,000 population). In Indonesia, the total number of premature deaths under BAU2030 would be 52,628 cases (26 per 100,000 population) while in Thailand it would be 13,420 cases (23 per 100,000 population). Spatially, the maximum value of 7 cases per 100,000 population per grid was seen in the populated areas over the eastern part of Java Island, Indonesia.

RED2030 would avoid a total number of premature deaths in the SEA countries of 401,000 cases (63 per 100,000 population) from that estimated for BAU2030. In Indonesia, the number of avoided premature deaths would be 103,448 (49 per 100,000 population) while in Thailand it would be around 21,235 (36 per 100,000 population). More than 50% the total number of avoided premature deaths in the SEA countries would be realized in Indonesia and Thailand, where the mitigation measures were considered. The values estimated in this study were lower than those by Shindell et al. (2012) for both Indonesia (74 per 100,000 population) and Thailand (68 per 100,000 population) (data extracted from http://www.giss.nasa.gov/staff/dshindell/Sci2012/FS5/). This is because our study considered a smaller set of emission reduction measures as compared to Shindell et al. (2012) who considered 7 BC reduction measures and 7 methane reduction measures in different sectors. In addition, the health effects in this study were quantified only for $PM_{2.5}$ while Shindell et al. (2012) also included effects of ground level ozone as well as other co-emitting species of BC.

The uncertainty of modeled $PM_{2.5}$ and BC was caused by several factors such as missing sources and other uncertainty in the EI data, incorporation of LRT, grid average to point-based observation and so on. In addition, the limited observation data available have prevented from a more comprehensive model performance evaluation. These all would be translated into the uncertainty of the health and BC DRF effect results. Even though we used

the change in the annual ambient PM$_{2.5}$ concentration ($\Delta C$) in the calculation of the health risk, the resulting impact of the intervention (emission reduction) may still contain a high uncertainty. Further, the regional and/or country specific CR data for PM$_{2.5}$ should be developed to improve the impact assessment of emission reduction scenarios on the premature mortality.

**3.2.2 BC direct radiative forcing**

Black carbon DRF in SEA is of interest to understand the impacts of BC emission on the climate warming. Therefore, BC AOD ($\tau_{ab}$) was calculated as the difference between the total AOD (scattering + absorbing) and scattering AOD, i.e. following the same method presented in the literature (Landi and Curci, 2011). In our accompanying paper (Permadi et al., 2017a), the results of BC AOD (using eq. 2) for the BY2007 were used for
the model evaluation without evaluation of BC. The monthly averages of BC DRF for the base year 2007 are presented in Figure 1 for selected months of August (wet season in the northern part of the modeling domain but dry season in the southern part of the domain) and January, February, December (dry season in the northern but wet season in the southern part of the modeling domain). Note that the BC DRF values represent the forcing at the top of the model layer and not that at the Earth surface. In most of the cases, higher BC DRF was seen in the
southern part of China but in the following discussion, we only focused on the results for the SEA countries, i.e. the SEA domain.

December had a vast area of high BC DRF in the SEA domain with the maximum monthly average value of 1.4 W m$^{-2}$ (seen over the Malaysian peninsular and western part of Java Island, Indonesia). The second highest monthly BC DRF was shown in February with a relative wide area of high values over the western part of
Malaysian peninsular and western part of Kalimantan Island, Indonesia. In August, the BC DRF values were lower over the SEA domain with the maximum monthly average seen over the border between Thailand and Myanmar and in Riau Province of Sumatera Island, Indonesia. January had the lowest BC DRF in all the presented months with the monthly maximum in the SEA domain of 0.9 W m$^{-2}$, seen over the western part of Malaysian peninsular and eastern part of Java Island, Indonesia. The influence of the emissions from the up-wind part of the
SEA domain was more pronounced in the months when the northeast monsoon was predominant (January, February and December). In these months, the highest monthly values were simulated outside the SEA domain, i.e. above the southern part of China and Taipei (maximum of 1.4 – 1.9 W m$^{-2}$).

The monthly distributions of BC DRF were consistent to the simulated monthly BC AOD. In this study we assumed the internally mixed state for BC when calculating AOD by AODEM and this may overestimate the light
absorption of BC (Jacobson, 2001). Our estimated BC DRF values are comparable to a previous study by Kim et al. (2012) who estimated the monthly BC DRF values for Korean peninsula of 1.2 – 1.5 W m$^{-2}$ but our results were higher than the global average BC DRF suggested by Jacobson (2000) of 0.55 W m$^{-2}$.

Note that there are several factors contributing to the uncertainty in estimating the BC DRF, such as underestimation of the LRT contribution due to the current model vertical set-up, missing sources and other
uncertainty in the EI data. Further, BC DRF was calculated for clear sky conditions therefore effect of BC on the cloud microphysics (Chýlek et al., 1996) was not incorporated.

The spatial distributions of the simulated annual average BC DRF of BY2007, BAU2030 and RED2030 scenarios are presented in Figure 4. In this section, we discussed only the values estimated for the SEA countries, i.e. excluding the high values occurred in the southern part of China. The maximum simulated BC DRF for the SEA countries under BAU2030 (2.0 W m$^{-2}$), appeared over Riau province, Sumatera Island of Indonesia, and this is about 1 W m$^{-2}$ higher than that in BY2007 (0.98 W m$^{-2}$). Under RED2030, the highest BC DRF would be 1.4 W m$^{-2}$ (appeared above Riau province, Indonesia), i.e. about 0.6 W m$^{-2}$ lower than that of BAU2030. In Thailand, the maximum BC DRF in 2007 was 0.40 W m$^{-2}$, appeared at Chonburi province and it increased by two times under BAU2030. Under RED2030, the maximum BC DRF in Thailand would be 0.48 W m$^{-2}$, appeared at the same place, i.e. only a slight increase as compared to BY2007.

Apart from Indonesia and Thailand for which the mitigation measures were simulated, other countries in the modeling domain following the RCP8.5 emission pathway also gained benefits because the pollutants (i.e. SLCPs) were also mitigated in this IPCC scenario (Riahi et al., 2011) but not the major GHGs (Table 1). Other parts of the modeling domain, i.e. the southern part of China and other SEA countries, also gained quite similar ranges of the BC DRF reduction and the health benefits as Indonesia and Thailand under RED2030. It is worth mentioning, as seen in Table 1, the BC emissions in the SEA domain were lower in RED2030 as compared to BY2007 but the AOD and BC DRF were higher than the respective BY2007 levels and these could be related to the emissions in other part of the domain and outside the modeling domain.

Our study thus demonstrated that the measures implemented to reduce BC (and PM) under RED2030 may bring in substantial benefits in avoiding the premature mortality and reduction in the BC DRF. Future studies should focus on the improvement of emission inventory data, modeling set-up, as well as the inclusion of the cloud microphysics in the radiative forcing calculation. The climate feedback should be expressed as the temperature changes in the future studies. The impacts on crop production and materials should also be considered and the monetary values of the benefits should be presented to better inform policy makers and to promote mitigation measures for the SLCPs.

## 4. Conclusions

This study is a continuation of our previous paper (Permadi et al., 2017a, focusing on the model performance evaluation for the BY2007) and presents the development of two emission scenarios for SEA in 2030, (BAU2030 and RED2030) to assess the associated impacts on the premature mortality and BC DRF in the region. BAU2030 assumed a linear increase of activity levels of the key anthropogenic sectors in Thailand and Indonesia, and featured a BC emission increase by 1.6 times in Thailand and 1.3 times in Indonesia as compared to BY2007. For other countries in the domain the projection was done using the average GDP growth rate, and the emission growth rate was 2.2 times of BY2007 for all species. RED2030 considered the emission reduction measures in four major anthropogenic sources for Indonesia and Thailand (road transport, residential combustion, industry, open burning), the two large contributors to the SEA emissions, and featured a cut in BC emission by 45% in Indonesia and 51% in Thailand as compared to BAU2030. The emissions reductions for Thailand and Indonesia under RED2030, calculated based on their national master plans, were quite close to those specified for these countries

under IPCC RCP2.6, hence suggested that by implementing the current policies the countries can contribute to the 2.0° target achievement.

With other countries in the modeling domain following the IPCC RCP8.5 pathway, the collective BC emissions from the domain under RED2030 would reduce by 16% from the BY2007 amount. For the SEA domain, the BAU2030 featured a total emission increase in the BC and $PM_{2.5}$ emissions by 1.6 times as compared to BY2007, respectively. Under RED2030, the emissions of $PM_{2.5}$ would reduce by 2.4 times and BC emission would reduce by 2.1 times from the respective BAU2030 levels.

WRF/CHIMERE/AODEM modelling system simulation results provided the PM ambient concentrations (i.e. $PM_{2.5}$, $PM_{10}$, and BC), AOD and BC DRF under different scenarios which showed substantial benefits of the emission reduction under RED2030 in improving regional air quality and BC DRF reduction. Under BAU2030, assuming "no intervention", the increase in annual $PM_{2.5}$ levels would induce an additional number of premature deaths of 30/100,000 population above the BY2007 in the SEA domain. The reduction measures implemented under RED2030 would help to cut down (avoided) the total number of the premature deaths by 63 per every 100,000 population as compared to BAU2030. For the two countries where specific measures were to be implemented, RED2030 would help to avoid 49 premature deaths in Indonesia and 36 in Thailand per every 100,000 population. RED2030 would also slow down the increase in the BC DRF over the SEA domain, i.e. lowering the maximum annual average BC DRF from 2.0 W $m^{-2}$ under BAU2030 to 1.4 W $m^{-2}$ under RED2030.

Future studies should assess the potential impacts of the emission reduction on agricultural crops (mainly via ozone formation) and this is important because agriculture is the major economic sector in SEA. Other pollutants (beside PM and BC) should be included in the assessment of health impacts. Likewise, for the long-term climate effects, the induced emission reduction of major GHGs by the measures should also be included. Multiyear simulations using an on-line coupled climate – chemistry modeling system should be conducted to provide a more realistic impact resulting from emission reduction scenarios on air quality and climate in SEA.

**Acknowledgments**

We acknowledged Dr. Didier Hauglustaine for sharing the simulation results of the LMDz/INCA used for the boundary conditions in this work. This research was financially supported by the French government under the Asian Institute of Technology (AIT)/Sustainable Development in the Context of Climate Change (SDCC) – France Network cooperation and by the United States Agency for International Development (USAID) under the PEER-SEA (Partnerships for Enhanced Engagement in Research for Southeast Asia) research project.

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

**Table captions**

**Table 1: Emissions of BY2007, BAU2030 and RED2030 for Indonesia, Thailand and other countries in the modeling domain.**

Table 2: Summary of emission reduction scenarios for the SEA domain.**

**Table 1: Emissions of BY2007, BAU2030 and RED2030 for Indonesia, Thailand and other countries in the modeling domain.**

| Emission (Gg y$^{-1}$) | SO$_2$ | NOx | NH$_3$ | PM$_{10}$ | PM$_{2.5}$ | BC | OC | NMVOC | CO | CO$_2$ | CH$_4$ | N$_2$O |
|---|---|---|---|---|---|---|---|---|---|---|---|---|
| **Indonesia** | | | | | | | | | | | | |
| BY2007[a] | 997 | 3,282 | 1,258 | 2,046 | 1,644 | 226 | 674 | 3,840 | 24,169 | 508,022 | 3,950 | 180 |
| BAU2030 | 1,944 | 5,251 | 2,214 | 2,844 | 2,252 | 305 | 903 | 5,760 | 29,003 | 810,413 | 5,530 | 324 |
| RED2030 | 1,785 | 4,923 | 2,151 | 859 | 904 | 167 | 506 | 4,608 | 17,885 | 378,193 | 3,792 | 157 |
| Ratio BAU2030/BY2007 | 1.95 | 1.6 | 1.76 | 1.39 | 1.37 | 1.35 | 1.34 | 1.5 | 1.2 | 1.5 | 1.4 | 1.8 |
| Ratio RED2030/BY2007 | 1.79 | 1.5 | 1.71 | 0.42 | 0.55 | 0.74 | 0.75 | 1.2 | 0.74 | 0.7 | 0.96 | 0.87 |
| Ratio RED2030/BAU2030 | 0.92 | 0.94 | 0.97 | 0.3 | 0.4 | 0.55 | 0.56 | 0.8 | 0.62 | 0.47 | 0.69 | 0.48 |
| **Thailand** | | | | | | | | | | | | |
| BY2007 | 827 | 701 | 469 | 782 | 607 | 47 | 240 | 1,120 | 9,095 | 260,988 | 1,053 | 84 |
| BAU2030 | 1,613 | 995 | 727 | 1,298 | 953 | 77 | 341 | 1,725 | 9,186 | 287,087 | 1,685 | 151 |
| RED2030 | 1,340 | 925 | 661 | 712 | 498 | 38 | 127 | 1,366 | 6,730 | 219,230 | 853 | 80 |
| Ratio BAU2030/BY2007 | 1.95 | 1.42 | 1.55 | 1.66 | 1.57 | 1.63 | 1.42 | 1.54 | 1.01 | 1.1 | 1.6 | 1.8 |
| Ratio RED2030/BY2007 | 1.62 | 1.32 | 1.41 | 0.91 | 0.82 | 0.8 | 0.53 | 1.22 | 0.74 | 0.84 | 0.81 | 0.95 |
| Ratio RED2030/BAU2030 | 0.83 | 0.93 | 0.91 | 0.55 | 0.52 | 0.49 | 0.37 | 0.79 | 0.73 | 0.76 | 0.51 | 0.53 |
| **Other countries in the modeling domain** | | | | | | | | | | | | |
| BY2007 | 8,940 | 6,886 | 3,900 | 5,522 | 4,184 | 528 | 1,287 | 10,416 | 55,308 | 2,291,381 | 29,186 | 677 |
| BAU2030[b] | 19,668 | 15,149 | 8,580 | 12,148 | 9,205 | 1,162 | 2,831 | 22,915 | 121,678 | 5,041,038 | 64,209 | 1,489 |
| RED2030[c] | 6,169 | 7,781 | 5,070 | 4,915 | 3,640 | 444 | 1,133 | 10,937 | 54,202 | 2,749,657 | 34,439 | 785 |
| Ratio BAU2030/BY2007 | 2.2 | 2.2 | 2.2 | 2.2 | 2.2 | 2.2 | 2.2 | 2.2 | 2.2 | 2.2 | 2.2 | 2.2 |
| Ratio RED2030/BY2007 | 0.69 | 1.13 | 1.3 | 0.89 | 0.87 | 0.84 | 0.88 | 1.05 | 0.98 | 1.2 | 1.18 | 1.16 |
| Ratio RED2030/BAU2030 | 0.31 | 0.51 | 0.59 | 0.41 | 0.4 | 0.38 | 0.4 | 0.48 | 0.45 | 0.55 | 0.54 | 0.53 |

Note: [a] Permadi et al., 2017b. [b] - GDP average growth rates from 2000 – 2007 for other countries were obtained from http://data.worldbank.org/indicator/NY.GDP.PCAP.KD.ZG. Population weighted average GDP for other SEA countries and southern part of China was calculated to construct BAU2030/BY2007 ratio. [c] -2030/2007 ratio was extracted from RCP8.5 pathways taken from (http://tntcat.iiasa.ac.at:8787/RcpDb/)

**Table 2: Summary of emission reduction scenarios for the SEA domain.**

| Emission scenario | Remarks | BY2007 | BAU2030 | RED2030 |
|---|---|---|---|---|
| Emission of $PM_{2.5}$ (Gg y$^{-1}$) | SEA domain | 3,171 | 5,230 | 2,203 |
| Emission of $PM_{10}$ (Gg y$^{-1}$) | SEA domain | 5,036 | 9,001 | 3,537 |
| Emission of BC (Gg y$^{-1}$) | SEA domain | 373 | 603 | 289 |
| $PM_{2.5}$ in SEA (µg m$^{-3}$) | Hourly maximum | 189 | 296 | 146 |
| | Highest annual average[a] | 32.0 | 36.4 | 21.1 |
| | Highest monthly average[b] | 82 | 97 | 58 |
| $PM_{10}$ in SEA (µg m$^{-3}$) | Hourly maximum | 327 | 472 | 247 |
| | Highest annual average[a] | 50 | 58 | 34 |
| | Highest monthly average[b] | 127 | 150 | 88 |
| BC in SEA (µg m$^{-3}$) | Hourly maximum | 39 | 59 | 32 |
| | Highest annual average[a] | 6.0 | 7.2 | 4.3 |
| | Highest monthly average[b] | 21 | 22 | 11 |
| BC AOD in SEA | Highest monthly average[b] | 0.08 | 0.24 | 0.11 |
| BC DRF in SEA (W m$^{-2}$) | Highest annual average[a] | 0.98 | 2 | 1.4 |
| Mortality cases per every 100,000 of population[c] | Total number of additional mortality cases in the SEA domain compared to BY2007 | | (+)30[d] | (-)63[e] |
| | Total number of additional mortality cases in Indonesia | | (+)26[d] | (-)49[e] |
| | Total number of additional mortality cases in Thailand | | (+)23[d] | (-)36[e] |

Note: this table does not include the values simulated for southern China part of the modeling domain total.
[a] Highest annual average value observed in the SEA domain, [b] Highest monthly average value observed in the SEA domain, and [c] Sum of all value in the SEA/country, (+) addition, and (-) reduction (avoided), [d] compared to BY2007, [e] compared to BAU2030.

**Figure captions**

**Figure 1: Spatial distribution of monthly average BC direct radiative forcing for the selected months, BY2007**

**Figure 2: Simulated annual average concentrations of BC and PM$_{2.5}$ for BY2007, BAU2030 and RED2030 in μg m$^{-3}$**

**Figure 3: The difference in the number of mortality cases (cases per 100,000 population) between BAU2030 and BY2007 (BAU2030-BY2007), and between BAU2030 and RED2030 (BAU2030-By2007)**

**Figure 4: Spatial distribution of annual average BC DRF under BY2007, BAU2030 and RED2030 scenarios.**

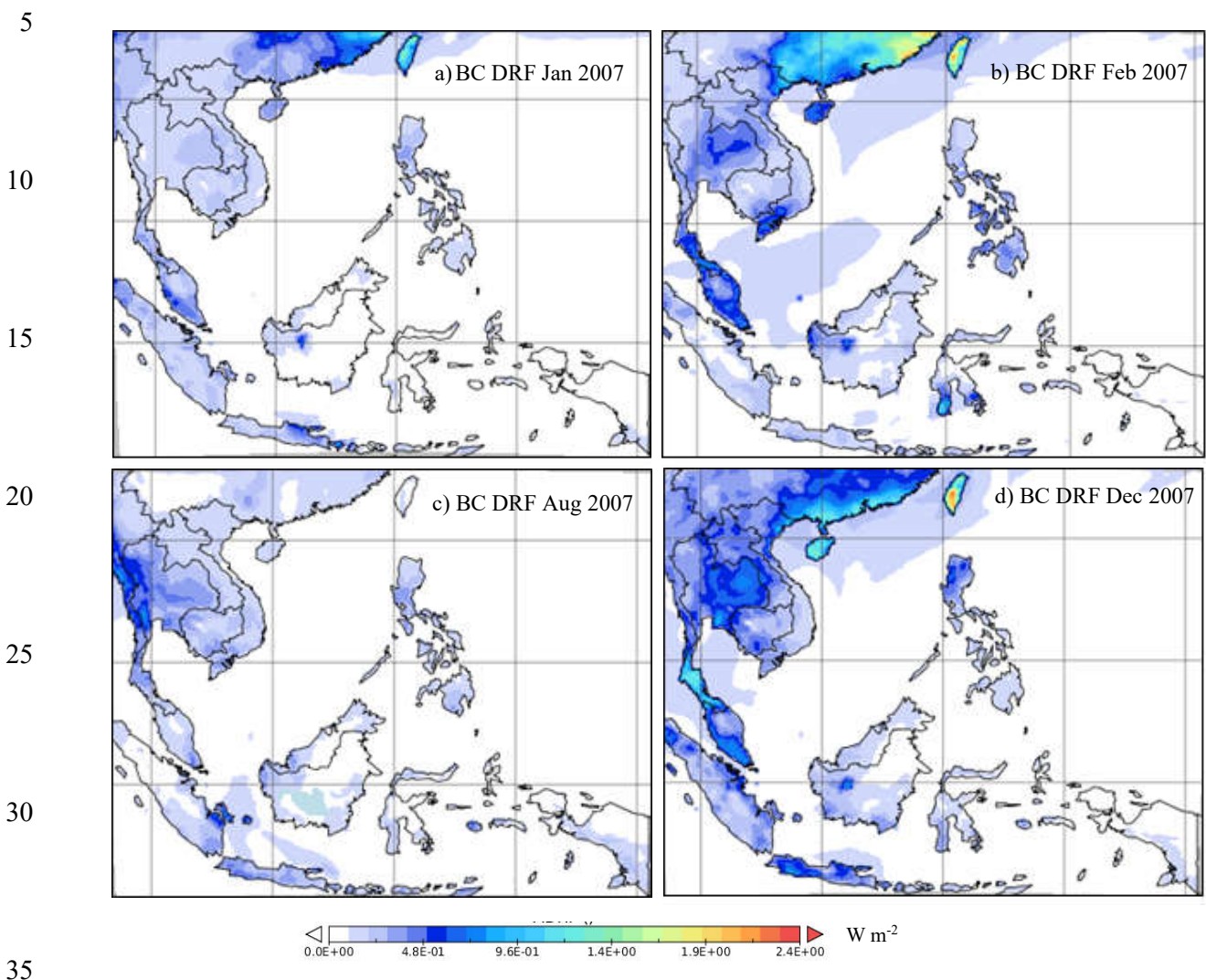




**Figure 1: Spatial distribution of monthly average BC direct radiative forcing for the selected months, BY2007**




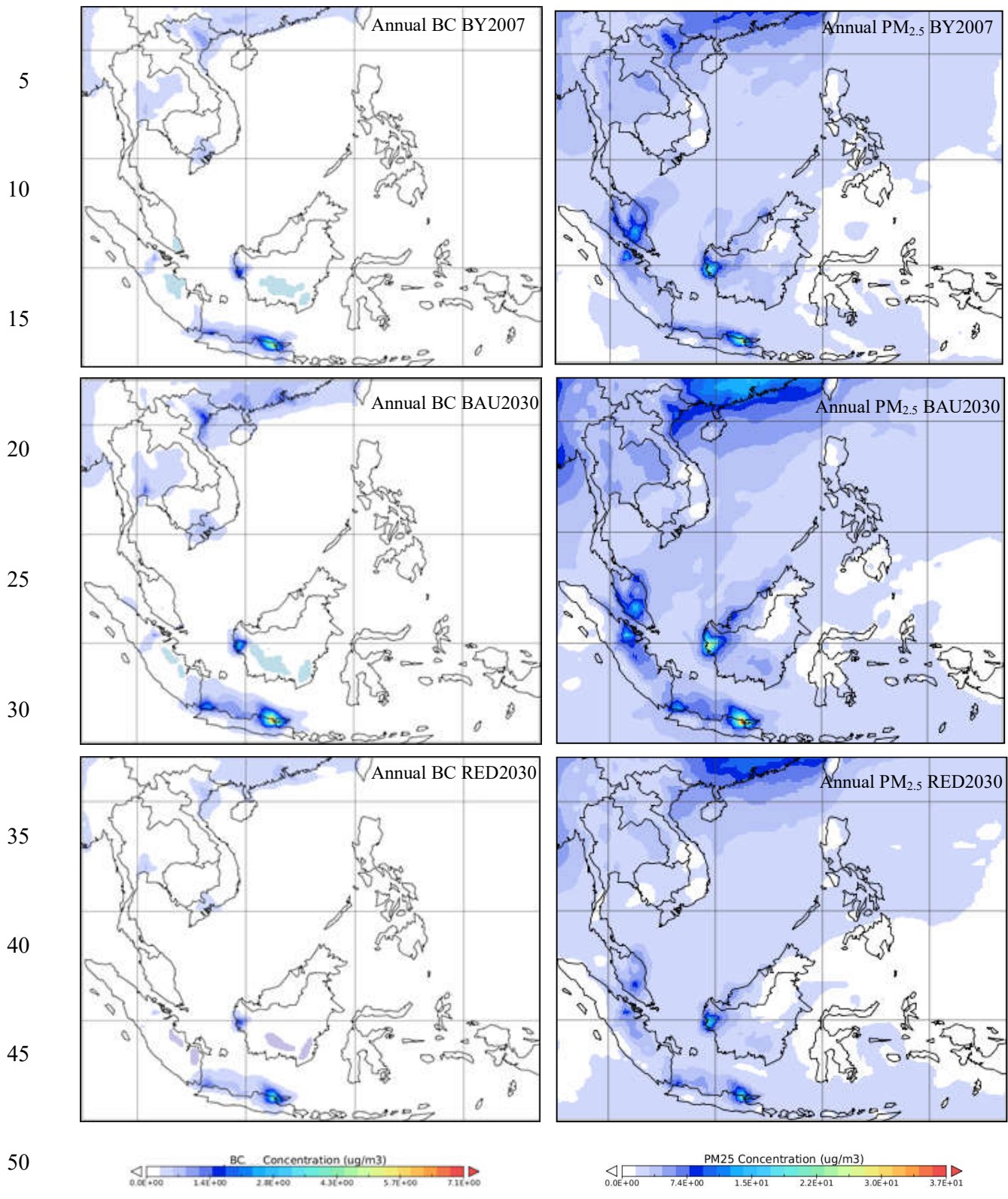

**Figure 2: Simulated annual average concentrations of BC and PM2.5 for BY2007, BAU2030 and RED2030 in μg m⁻³**

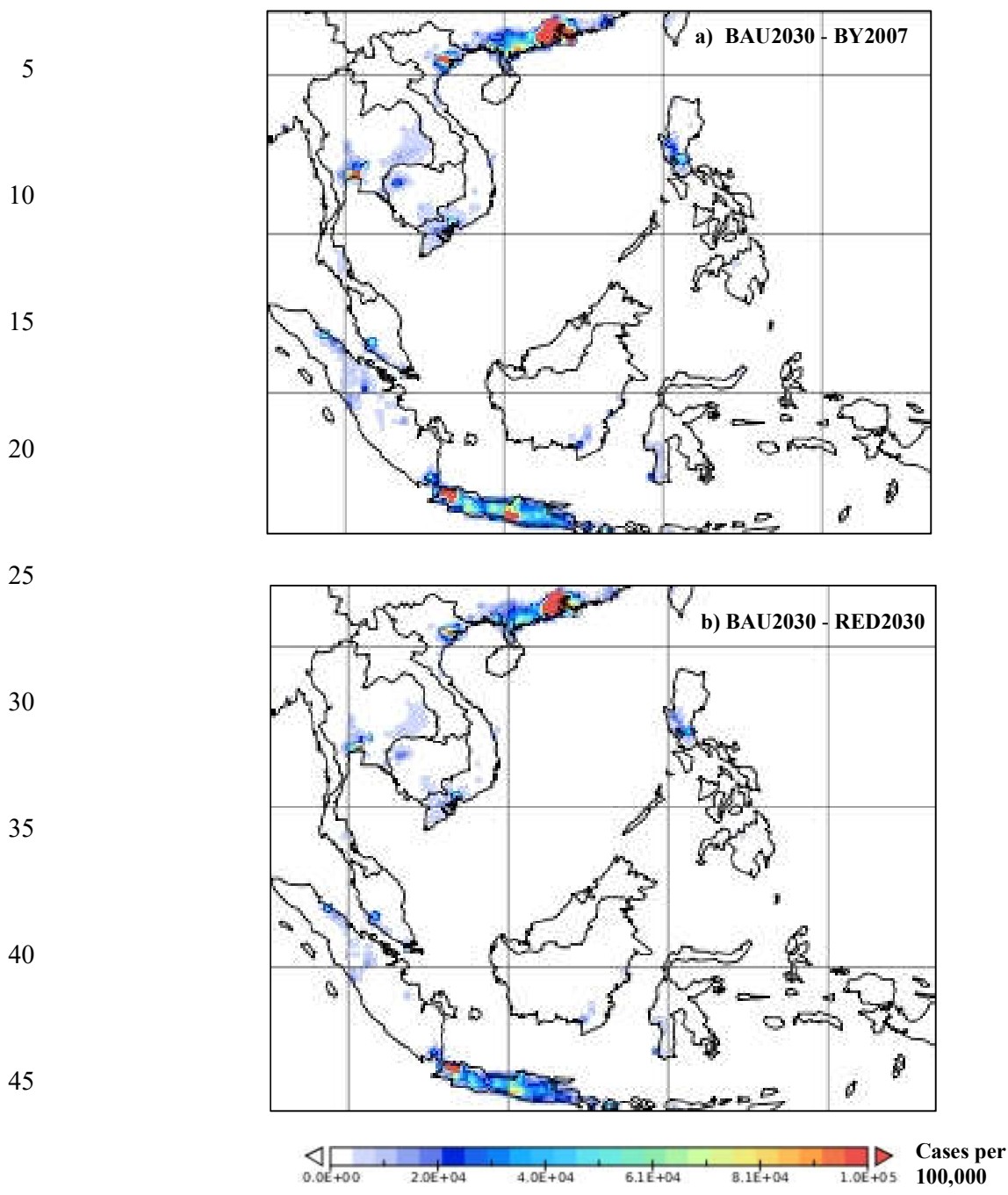

**Figure 3: The difference in the number of mortality cases (cases per 100,000 population) between BAU2030 and BY2007 (BAU2030-BY2007), and between BAU2030 and RED2030 (BAU2030-By2007).**

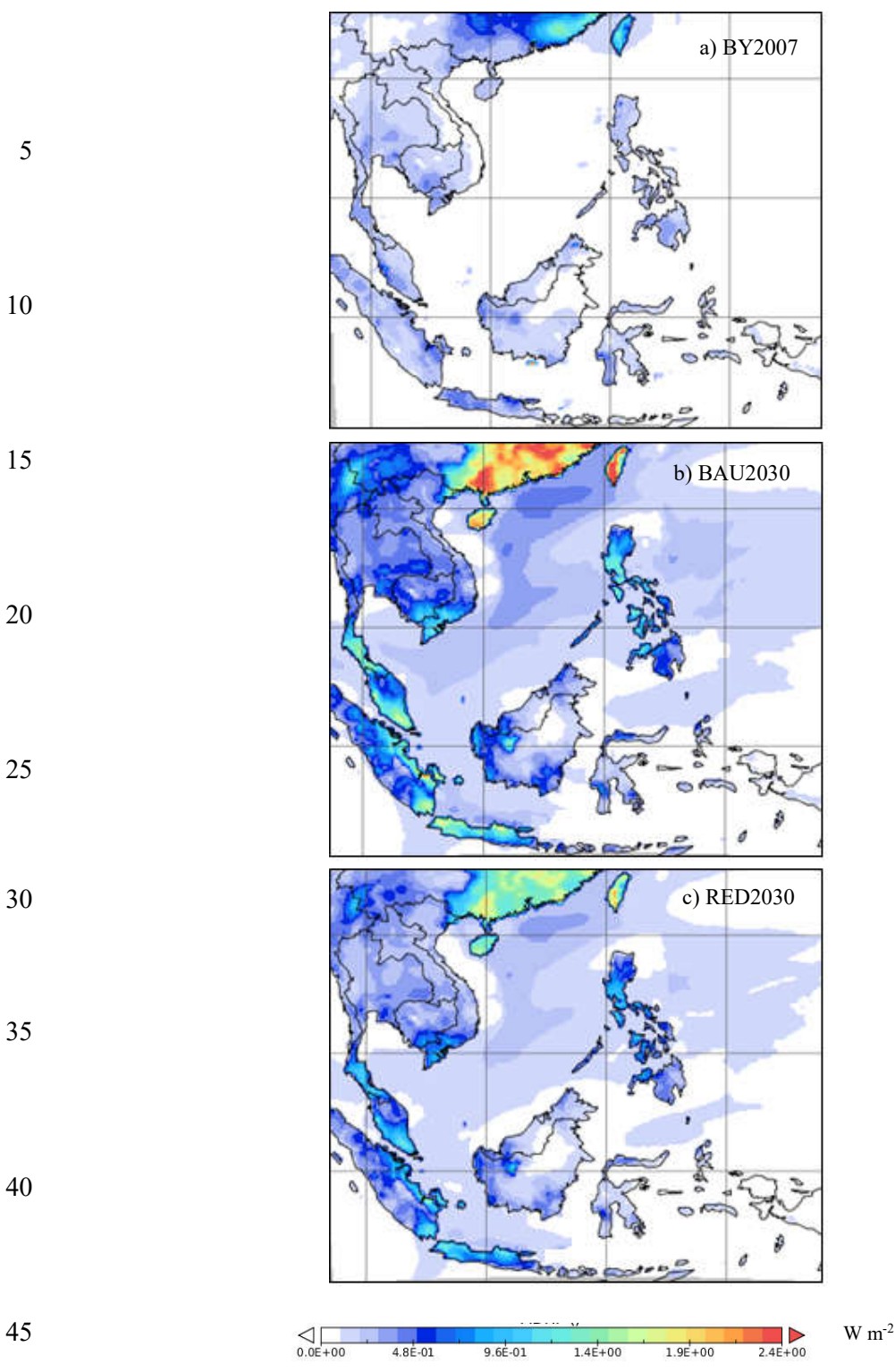

**Figure 4: Spatial distribution of annual average BC DRF under BY2007, BAU2030 and RED2030 scenarios.**