# Peer review of "Assessment of emission scenarios for 2030 and impacts of black carbon emission reduction measures on air quality and radiative forcing in Southeast Asia"

_Atmospheric Chemistry and Physics, 2017_

## Referee Comment (RC1) · Anonymous Referee #1 · 18 Jun 2017

This is the second part of the two parts paper on the BC emission and its impact on air pollution in Southeast Asia. In this manuscript, the authors examined the impact of different scenarios of BC emission on air pollution and climate. However, the meteorological fields are fixed with year 2007. Therefore, air pollution and aerosol radiative forcing as well as the number of premature death are largely proportion to the levels of BC emissions. I found this paper doesn't add any new insight to the problem. So, I do not recommend this paper for publication.

[Figure]

Specific Comments:

1) Experiment setting is not clearly described. If I understood correctly, all meteorological variables and SST are same as year-2007 (lines 8-9 on page 3). Only emissions and chemistry boundary conditions are changed. This modeling experiment doesn't represent the future scenarios. 2) Is DRF for all-sky or clear-sky? 3) Lines 24-26 on page 3: I understand that long-term data record is not available for many cities, but 4-19 years are too short to determine trend or BAU scenario. 4) Line 34 on page 10, "bring in substantial benefit to human health and climate": Here, by the benefit to climate, the author seem to indicate the reduction of BC DRF due to reduction of BC emission. First of all, could you clarify whether DRF is for all sky or clear sky? Also, the reduction does not necessarily mean beneficial. How to define beneficial for this case, climate? Also, it is not climate, but climate forcing. 5) Figure S1: Dotted line (=ratio of 1.0) does not match with color bars. For example, dotted line in the maritime continent is not same as that along the east coast of India.

---

## Referee Comment (RC2) · Anonymous Referee #2 · 11 Sep 2017

The study investigates the co-benefits of black carbon and PM2.5 emission reductions from climate and health related perspectives. Part 1 summaries the development of South East Asia (SEA) emission inventory, and validation of air quality model, while part 2 summaries the climate and health benefits from the reductions of BC and PM2.5 in 2030 for SEA. Overall, the paper is written within the scope of ACP and in a reasonable quality. Major changes, rerun and more in-depth explanation should be done before accepting for ACP.

[Figure]

General comments: 1) The experimental plan was well established. However, the model implementation (particularly in the air quality part) was not well delivered. It is necessary to redo the simulation to resolve that.

For example (described in part 1), CHIMERE only uses 8 vertical layers (up to 5.5 km). How the authors can consider long-range transport of air pollutants (e.g., PM2.5 and BC), which often happens at the free troposphere (from 2 km to 16 km – tropic). How the authors come up AOD only with 5.5 km column? Although I understand that the vertical profile of aerosol concentration goes down as increase of height. However, AOD is measured based on the entire vertical column. The influence of missing mid- and top- tropospheric (i.e., about 10 km) concentration could be huge, even with low aerosol concentrations. This could introduce a huge bias to the results. Moreover, major biomass burning influence occurs at the free troposphere, as plume-rise normally bought biomass burning BC and PM2.5 up to 6-10km. Without upper layer information, I am not sure how the authors can consider that. I agree with the authors that for the anthropogenic emissions, the influence of upper layer may not have much impact to the surface. That is only limited to the case with "no long-range transport", "no biomass burning", and "no tropospheric downwash.

2) Be honest, model performance is very bad for 30 km x 30 km resolution. For example, in part 1 Table 3, the RMSE for some of the sites is more than 10 degree Celsius. The highest could reach 20.5 degree Celsius. I am not sure whether I can trust these results. Please double check the statistic results.

3) In order to understand the part 2, more description of the regridding process, and what spatial interpolation methods (e.g., spatial surrogate) used in the study for generating the model emissions are important, as the countries in marine time continents are widely spread. How authors use those regional emissions from 50 km to 1 degree emission results (e.g., EDGAR or CGRED) to derive 30 km x 30 km resolution for the simulation.

4) The information from LMDZ/INCA was insufficient. I can't find enough information to understand the process. Any species mapping table, or cross-referencing table is available.

Specific comments 1) P1. L 21, more information on LMDZ/INCA. 2) P2. L21-23, please rewrite the sentence, I don't fully understand. 3) P3. L7, why only 1 year of WRF? Is 2007 a typical average year? Is it a wet or dry year? Normally, I will do three years simulation for any climate related study since it has to take into consideration of the interannual variability. Of course, this study relates more on air quality. I can understand using 1 year data. However, the authors may need to demonstrate 2007 is an average typical year, in terms of temperature, and precipitation. Sometimes, ENSO effect may have a huge impact, and may bias the results in the marine-time continents. 4) P3. L13, global LMDZ/INCAS has 19 layers, how the authors interpolate the results into 8 layers CHIMERE results. Is the 2030 case using 2030 boundary conditions from LMDZ/INCAS. The part 1 paper didn't mention anything related 2030 scenarios for LMDZ model. More information should be provided. 5) Moreover, for the Figure S1, the description should be 2030/2007, not 2030/2006. Also, which case is it for 2030? BAU, RED? Very unclear. Annual concentration or monthly average (Jan, Aug)? 6) P3. L31, Yes, same EFs for 2007 were used for 2030 may contribute a certain uncertainty to the projected emission. Will cause increase or decrease? Can the authors provide more explanation? 7) P3. L31, Is the projection align well with IPCC projection for 2030 for those local projections? 8) P6. L34, recent years, Thailand has started restrict local burning. How this may affect the projection? 9) P8. L25, as mentioned from the general comments, only 8 layers (up to 5.5km) may not cover the entire vertical profile. What may be the impact on this? Also, from the part 1 of the paper, the BC was well underestimated (Figure 8 in part 1 using AERONET data). How this underestimations of BC and PM2.5 would affect the results on the analysis of direct radiative forcing? 10) P9. L19. The authors mentioned the different between BY2007 and 2030 are listed 1.2, 2.4 and 4.3 ug/m3. These results seem falling into the uncertainty range of the results. As shown in Figure 2 to 5 in the part 1, the modeling errors of BC and PM2.5

are huge (1-5 times lower than observed). Can the authors elaborate more on that? How this underestimations may influence the results on health impact analysis.

In P5 L13, "CR data was obtained from Smith who indicates that every increase in PM2.5 by 1 ug/m3 is approximately associated with an 1.006% increase in the risk." How the model uncertainty affects the risk calculation?

11) Table 1. Header for PM1. And PM2. Were not showing properly. 12) Table 2. Co-benefits of emission reduction? What kind of co-benefits? I think the title should be "summary of emission reduction scenarios for the SEA domain". The value of "327 and 472 ug/m3 for hourly maximum seems to be very large. Please double check. 13) Figure 3. Very strange to see areas outside of Jakartar would have the same impacts as Jackartar. As shown in Figure 2, ] high concentrations of PM2.5 and BC are found in Jakartar, not other places in the island. However, the mortality cases in Figure 3 are all red for the island.

---

## Author Comment (AC1) · 28 Oct 2017

This is the second part of the two parts paper on the BC emission and its impact on air pollution in Southeast Asia. In this manuscript, the authors examined the impact of different scenarios of BC emission on air pollution and climate. However, the meteorological fields are fixed with year 2007. Therefore, air pollution and aerosol radiative

forcing as well as the number of premature death are largely proportion to the levels of BC emissions. I found this paper doesn't add any new insight to the problem. So, I do not recommend this paper for publication.

Response: Thank you for your comment. We agree that the approach is standard but we also argue that our key contribution in this part was to develop the emission scenarios taking into account the country specific data (BAU2030) and relevant policies for future emission (RED2030, emission reduction) hence could capture the realistic development of the region. We did not rely on the IPCC method in development of the emission scenarios but when we compared the emission scenarios results with the IPCC we found that the RED2030 is very much aligning with the IPCC RCP2.6 hence suggesting that the current master plans in the considered countries (Thailand and Indonesia) could lead to achieving 2.0 degree target. It is detailed in the revised manuscript lines 34-37, page 7: "As compared to BY2007, BAU2030 increased the regional BC and CO2 emissions in 2030 by nearly 1.89 and 1.41 times and were well above the increase rate under Intergovernmental Panel on Climate Change (IPCC) RCP6.0 for BC and O2 of 1.03 and 1.23 times for SEA".

And lines 34-37 and lines 1-2, page 8: "Our national BC RED2030 to BY2007 emission ratios for Indonesia (0.74) and Thailand (0.81) were close to the IPCC RCP2.6 scenario for the 2 countries of 0.82 and 0.81, respectively. For CO2, the emission ratios were 0.7 and 0.84 which were slightly lower than the IPCC RCP2.6 values of 0.92 and 0.86, respectively. It therefore shows that the RED2030 is very much aligning with the IPCC RCP2.6 hence suggesting that the current master plans in the considered countries (Thailand and Indonesia) could lead to achieving 2.0° target".

This study, to our best knowledge is the first that use modeling to analyze the co-benefits at the SEA regional scale. The results therefore would contribute scientific evidence to promote the co-benefit approach that is currently not incorporated in the policy in any country in the region. We have now highlighted this point in our revised MS, lines 1-4, page 3: "To our best knowledge, this is the first study addressing cobenefit for the SEA region hence the results would contribute scientific evidences to promote the co-benefit approach that is currently not incorporated in the policy in any country in the region". We also agree that it would be interesting to use the future meteorology for the 2030 simulation. However, we selected to use 2007 meteorology because the WRF performance was shown acceptable in Part I (now titled as "Integrated emission inventory and modeling to assess distribution of particulate matter mass and black carbon composition in Southeast Asia"). It is therefore would be useful to check the impact of the emission reduction alone on the climate forcing and health effects in SEA using the WRF evaluated results. Our focus was to examine the impact of emission reduction measures hence we selected to use the tested meteorology (fixed on 2007) for the SEA. Nevertheless, we incorporated the future (2030) boundary conditions from the global CTM simulation of LMDz-INCA to account for the future inflows of pollutants from outside the domain. We recognize the importance of the incorporation of the future meteorology and multiyear simulations hence added this as a recommendation in the revised MS lines 11-13, page 13: "Multiyear simulations using the on-line coupled climate – chemistry modeling system should be conducted to provide a more realistic picture of the impacts resulting from emission reduction scenarios on air quality and climate in SEA".

Specific Comments: 1) Experiment setting is not clearly described. If I understood correctly, all meteorological variables and SST are same as year-2007 (lines 8-9 on page 3). Only emissions and chemistry boundary conditions are changed. This modeling experiment doesn't represent the future scenarios.

Response: Thank you. As noted above, we kept all meteorological variables and SST the same of 2007 to test the impacts of the emission changes in SEA. We agree that our experiment focused on the emission scenarios.

2) Is DRF for all-sky or clear-sky?

Response: The DRF was calculated for clear-sky. The sentence was revised to add

the clear-sky term in the revised MS in line 9, page 6: "To estimate clear sky BC DRF, a radiative transfer equation (eq. 2) was used following the approach presented......". More on the uncertainty due to the use of clear sky DRF is also discussed in response to comment #4 below.

3) Lines 24-26 on page 3: I understand that long-term data record is not available for many cities, but 4-19 years are too short to determine trend or BAU scenario.

Response: We realized the short period would introduce bias for the Business as Usual (BAU) scenario. However, we are limited with the availability of data for the two considered countries (Indonesia and Thailand). In addition, in 1997-1998 the region has gone through the drastic economic recession hence inclusion of the data before 1999 may introduce more bias also in the projection. Therefore, even when longer data period was available we still relied on the trend obtained for the period 1999-2007 for the sectors. Exception was for the petroleum refinery and mining production in Thailand that had only a short period of historical data (4 years) hence it may contribute to the uncertainty of the BAU scenario development. Luckily, these sectors do not contribute significantly to the total emissions.

This explanation is now added in the revised MS lines 2-8 page 4: "In principle, it is desirable to have long historical data periods for the emission projection (for Thailand and Indonesia). In addition, in 1997-1998 the region went through a drastic economic recession hence inclusion of the data before 1999 may induce a large bias in the projection. Therefore, even when a longer data period available for a sector we still relied on the trend obtained for the period of 1999-2007. Exception was for the petroleum refinery and mining production in Thailand that had only a short period of historical data (4 years) hence it may contribute to the uncertainty of the BAU scenario development. Nevertheless, these sectors do not contribute significantly to the total emissions". Footnote has been also added in the Table S2: "Note: only the 1999-2007 data period was used for the regression analysis. The data available before that was not incorporated because of the drastic economic recession in SEA during 1997-1998".

4) Line 34 on page 10, "bring in substantial benefit to human health and climate": Here, by the benefit to climate, the author seem to indicate the reduction of BC DRF due to reduction of BC emission. First of all, could you clarify whether DRF is for all sky or clear sky? Also, the reduction does not necessarily mean beneficial. How to define beneficial for this case, climate? Also, it is not climate, but climate forcing.

Response: The DRF was for clear sky only as also in the response to the previous comment #2. We agree that the calculation for clear sky only may add uncertainty to the results. We added this in the revised MS lines 12-16, page 12, and also changes to "climate forcing reduction, as per your suggestion.

"Our study thus clearly demonstrated that the measures implemented to reduce BC (and PM) under RED2030 may bring in substantial benefit to avoid premature mortality and reduction in the BC climate forcing (DRF). Note that, BC DRF was calculated for clear sky conditions therefore effect of BC on the cloud microphysics (Chálek et al., 1996) was not incorporated. Climate feedback in temperature changes should be quantified in the future study".

5) Figure S1: Dotted line (=ratio of 1.0) does not match with color bars. For example, dotted line in the maritime continent is not same as that along the east coast of India.

Response: Thank you for your comment. There was a plotting problem. We now removed the dotted contour lines for BC and OC in Figure S1 as it is already explained in the color scale.   Anonymous Referee #2

The study investigates the co-benefits of black carbon and PM2.5 emission reductions from climate and health related perspectives. Part 1 summaries the development of South East Asia (SEA) emission inventory, and validation of air quality model, while part 2 summaries the climate and health benefits from the reductions of BC and PM2.5 in 2030 for SEA. Overall, the paper is written within the scope of ACP and in a reasonable quality. Major changes, rerun and more in-depth explanation should be

done before accepting for ACP.

Response: Thank you for your comments and suggestions.

General comments: 1) The experimental plan was well established. However, the model implementation (particularly in the air quality part) was not well delivered. It is necessary to redo the simulation to resolve that. For example (described in part 1), CHIMERE only uses 8 vertical layers (up to 5.5 km). How the authors can consider long-range transport of air pollutants (e.g., PM2.5 and BC), which often happens at the free troposphere (from 2 km to 16 km – tropic). How the authors come up AOD only with 5.5 km column? Although I understand that the vertical profile of aerosol concentration goes down as increase of height. However, AOD is measured based on the entire vertical column. The influence of missing mid and top- tropospheric (i.e., about 10 km) concentration could be huge, even with low aerosol concentrations. This could introduce a huge bias to the results. Moreover, major biomass burning influence occurs at the free troposphere, as plume-rise normally bought biomass burning BC and PM2.5 up to 6-10km. Without upper layer information, I am not sure how the authors can consider that. I agree with the authors that for the anthropogenic emissions, the influence of upper layer may not have much impact to the surface. That is only limited to the case with "no long-range transport", "no biomass burning", and "no tropospheric downwash.

Response: Thank you for your input and we agree that, with the current vertical set-up, the long-range transport (LRT) that takes place in the free tropospheric layer may not be captured hence causing bias in the total column AOD and DRF results when LRT is substantial. However, because of the Planetary Boundary Layer (PBL) in our domain showed the maximum hourly value of 3,900 m, i.e. below the upper level of the domain (5,500 km), hence the mixing down effects of the LRT pollution from above PBL may be partly included. This has been explained in our accompanying paper (Part 1, now titled as "Integrated emission inventory and modeling to assess distribution of particulate matter mass and black carbon composition in Southeast Asia").

We used the previous version of the model (CHIMERE 2008c) with a simplified photolysis rate calculation (Madronich et al., 1998) that assumed the model domain is below the cloud, hence putting a constraint on model top to be maximum at 500 hPa. In our accompanying paper (Part I), we recommended for the future studies to use the most recent version of the model with updated radiative transfer model hence can compute photolysis reaction rates for extending the model domain vertically beyond 500 hPa. Regarding the biomass open burning, it was found that the aerosol extinction coefficient (in the transboundary haze affected sites in SEA observed by the level 2 NASA Cloud Aerosol Lidar with Orthogonal Polarization (CALIOP)) was not significant above the height of 5 km (Campbell et al., 2013). Thus, the uncertainty caused by the vertical model set-up is expected but the effect may be small.

We have added a discussion in the revised MS in lines 19-25, page 6 "Note that, the current vertical model set-up was taking 500 hPa as the model top hence the simulation results may not be able to capture the long-range transport (LRT) that took place in the free tropospheric layer. This may cause underestimation of the simulated results for the total column BC AOD and BC DRF. However, the biomass open burning emission of importance in the region, in particular, may not have significant effects on the aerosol extinction coefficient above the height of 5 km as shown by the CALIOP observations discussed in Campbell et al. (2013). Therefore, the uncertainty caused by the vertical model set-up but the effects on the simulation results may not be significant".

2) Be honest, model performance is very bad for 30 km x 30 km resolution. For example, in part 1 Table 3, the RMSE for some of the sites is more than 10 degree Celsius. The highest could reach 20.5 degree Celsius. I am not sure whether I can trust these results. Please double check the statistic results.

Response: Thank you for your correction. We checked the results and the situation was for the 2 stations in the Philippines that have large missing data and these were taken as "zero" in the statistical measure calculation (e.g. RMSE) template used. In the revised version of Part I we recalculated the statistical measures using only the hours

with observation data and results (Table 3, Part I) showed better performance. We also referred to Part I in the discussion.

3) In order to understand the part 2, more description of the regridding process, and what spatial interpolation methods (e.g., spatial surrogate) used in the study for generating the model emissions are important, as the countries in marine time continents are widely spread. How authors use those regional emissions from 50 km to 1 degree emission results (e.g., EDGAR or CGRED) to derive 30 km x 30 km resolution for the simulation.

Response: The emissions from EDGAR and CGRER are available at 0.5° x 0.5° (50 x 50 km2) resolution. The EI we prepared for the three countries had a resolution of 30 x 30 km2. We used the spatial analysis of "grid conversion" in Geographical Information System (GIS) to overlay the emission with the land mask map (generated from WRF, resolution of 30 x 30 km2) to specially regrid the border between land and sea. We added information in the revised MS lines 13-17, page 4: "The BY2007 emission data for those countries was taken from the international databases that were provided at a different resolution (i.e. 0.5° x 0.5°). Therefore, the spatial analysis of "grid conversion" in Geographical Information System (GIS) was used to convert the emission data in to the same grid resolution 30 x 30 km2 for the whole domain. The land mask map that was generated from WRF model with a resolution of 30 x 30 km2 was used to regrid the emissions in the border between land and sea by assigning zero emissions to the sea area".

4) The information from LMDZ/INCA was insufficient. I can't find enough information to understand the process. Any species mapping table, or cross-referencing table is available.

SpecifiĄc comments 1) P1. L 21, more information on LMDZ/INCA.

Response: Information on the LMDz/INCA model was added in the revised MS in lines 11-15, page 3: "However, the future simulations used the chemistry boundary conditions obtained from global chemistry–aerosol– climate model that couple online the LMDz (Laboratoire de Météorologie Dynamique, version 4) general circulation model (Hourdin et al., 2006) and the INCA (INteraction with Chemistry and Aerosols, version 3) model (Hauglustaine et al., 2014) using the global RCP8.5 emissions (Hauglustaine, 2013)". And lines 18, page 3: "The horizontal resolution is 1.90° in latitude and 3.75° in longitude".

However, since we obtained the results from the simulation conducted by Dr. Didier Hauglustaine (CNRS-LSCE) who was acknowledged in the MS, we did not present the results of LMDz/INCA in detail but added references for the publications that describe the model detail configurations.

P2. L21-23, please rewrite the sentence, I don't fully understand.

Response: The sentence has been re-written as follows (lines 28-30, page 2): "The results of this study could be useful for policy makers to provide information on the efficacy of different emission reduction measures and associated co-benefits on air quality, health, and climate forcing in SEA".

2) P3. L7, why only 1 year of WRF? Is 2007 a typical average year? Is it a wet or dry year? Normally, I will do three years simulation for any climate related study since it has to take into consideration of the interannual variability. Of course, this study relates more on air quality. I can understand using 1 year data. However, the authors may need to demonstrate 2007 is an average typical year, in terms of temperature, and precipitation. Sometimes, ENSO effect may have a huge impact, and may bias the results in the marine-time continents.

Response: We focused on the air pollutants that have short lifetime in the atmosphere, i.e. BC that is a short-lived climate forcing pollutant (SLCPs), hence considered only 1 year simulation. However we also recognize that longer term with multiyear simulation would provide more representative results. Therefore we added a recommendation in this MS (also to reflect the reviewer 1 comment) in lines 11-13, page 13: "Multiyear

simulations using the on-line coupled climate – chemistry modeling system should be conducted to provide a more realistic picture of the impacts resulting from emission reduction scenarios on air quality and climate in SEA".

The year of 2007 was selected because it was not affected by the strong El Niño and La Niña. The Oceanic Niño Index (ONI) analyses found that the year 2007 was categorized as weak La Niña (http://www.cpc.noaa.gov/products/analysis_monitoring/ensostuff/ensoyears.shtml). The information has been added in the revised manuscript in lines 10-11, page 3: "The base year of 2007 was selected because it was not affected by the strong El Niño and La Niña events (http://www.cpc.noaa.gov/products/analysis_monitoring/ensostuff/ensoyears.shtml)".

3) P3. L13, global LMDZ/INCAS has 19 layers, how the authors interpolate the results into 8 layers CHIMERE results. Is the 2030 case using 2030 boundary conditions from LMDZ/INCAS. The part 1 paper didn't mention anything related 2030 scenarios for LMDZ model. More information should be provided.

Response: The script of prep_bound.f90 (one routine in CHIMERE) read the monthly concentrations from the global CTM of LMDz/INCA and get the information of the lateral (lat_con) and top boundary (top_con) concentrations. The vertical profile of the concentration was interpolated to the similar layer (8 layers) set in the CHIMERE simulation. The results for 7 layers in the stratosphere (included in the LMDz/INCA simulation results) were not included in the boundary condition processing. More information was added in the revised manuscript for the 2030 simulation of LMDz in lines 18-20, page 3: "Boundary conditions from the LMDz/INCA were processed using the available routine in CHIMERE to read the monthly concentrations and get the information of the lateral (lat_con) and top boundary (top_con) concentrations".

In Part 1 paper, we focused on the model performance evaluation for the base year of 2007 hence not mentioned about scenarios which are the focus of this paper actually.

Now we added a sentence in the conclusions introducing this paper in lines 21-23, page 12: "This study is a continuation of our previous paper (Permadi et al., 2017a) which focused on the model performance evaluation for the BY2007. This paper focused on the development of two emission scenarios for SEA in 2030, BAU2030 and RED2030, and associated impact assessment on premature mortality and climate forcing".

5) Moreover, for the Figure S1, the description should be 2030/2007, not 2030/2006. Also, which case is it for 2030? BAU, RED? Very unclear. Annual concentration or monthly average (Jan, Aug)?

Response: Thank you for the correction. It was a typo, we corrected the table footnote to be 2030/2007 and the figure caption of "Annual average..." has been corrected to "Monthly average...." In Figure S1. The figure was modified and now only presents the ratios between simulated BC (and OC) for 2007 and 2030 by the LMDz/INCA (RCP8.5) that were used in our study for estimating the boundary conditions. The LMDz/INCA global simulation for RCP8.5 (including SEA) was used for the extraction of the boundary conditions. We used the ratio of 2030/2007 produced by LMDz/INCA for each species to scale up the boundary conditions from 2007 to 2030.

An explanation has been added in the revised MS lines 23-26, page 3: "Note that LMDz/INCA global simulation results for RCP8.5 (including SEA) were used for the estimation of the boundary conditions for our SEA domain. We used the ratio of simulated levels between 2030 and 2007 for each species to estimate the boundary conditions in 2030".

6) P3. L31, Yes, same EFs for 2007 were used for 2030 may contribute a certain uncertainty to the projected emission. Will cause increase or decrease? Can the authors provide more explanation?

Response: Thank you for the input. We agree that using the same emission factors contributes to the uncertainty of the projected emissions. In general, we expect the emission factors to be reduced when new technologies are available in 2030. However,

if the age limit is not enforced for vehicles then aged engine would have higher EF in the future. An explanation was added in the revised manuscript lines 10-12, page 4: "In general, we expect the EFs to be reduced in the future with the advance technology intrusion. Nevertheless, if the age limit is not strictly enforced for vehicles then aged engines would have higher EF in the future".

7)P3. L31, Is the projection align well with IPCC projection for 2030 for those local projections?

Response: We did not follow the IPCC Representative Concentration Pathway (RCP) scenarios in developing the scenarios for Indonesia and Thailand hence we do not expect 100% aligning. We found that under the IPCC RCP8.5 scenario for SEA domain, found a reduction in BC emission by 3.12% as compared with 2007 while under IPCC RCP6.0 an increase by 2.7% (1.027 times) was shown.

As compared to 2007, our BAU2030 increased the regional BC and CO2 emissions in 2030 and were well above the increase rate under IPCC RCP6.0 for SEA domain. We added this comparison in the revised MS, lines 34-37, page 7: "As compared to BY2007, BAU2030 increased the regional BC and CO2 emissions in 2030 by nearly 1.89 and 1.41 times and were well above the increase rate under Intergovernmental Panel on Climate Change (IPCC) RCP6.0 for BC and CO2 of 1.03 and 1.23 times for SEA".

Our national BC and CO2 emission ratios (RED2030/BY2007) were close to the RCP2.6 scenario and discussion has been added in the revised MS in lines 34-37, page 8 and lines 1-2, page 9: "Our national BC RED2030 to BY2007 emission ratios for Indonesia (0.74) and Thailand (0.81) were close to the IPCC RCP2.6 scenario for the 2 countries of 0.82 and 0.81, respectively. For CO2, the emission ratios were 0.7 and 0.84 which were slightly lower than the IPCC RCP2.6 values of 0.92 and 0.86, respectively. It therefore shows that the RED2030 is very much aligning with the IPCC RCP2.6 hence suggesting that the current master plans in the considered countries

(Thailand and Indonesia) could lead to achieving 2.0° target".

8) P6. L34, recent years, Thailand has started restrict local burning. How this may affect the projection?

Response: We included the scenario of the policy of Thailand government to ban the open burning in the RED2030 by reviewing the National Masterplan to Control Open Burning (MS, lines 32-36, page 4). We took into account the measure to limit the forest area burned not over 48,000 ha/yr which has been included in the manuscript (Table S1).

9) P8. L25, as mentioned from the general comments, only 8 layers (up to 5.5km) may not cover the entire vertical profile. What may be the impact on this? Also, from the part 1 of the paper, the BC was well underestimated (Figure 8 in part 1 using AERONET data). How this underestimations of BC and PM2.5 would affect the results on the analysis of direct radiative forcing?

Response: We realize the underestimation of PM and BC and that may be caused by several factors: 1) model vertical set-up, 2) uncertainty (e.g. missing sources) in EI, and 3) meteorological parameters, as well as gridded averaging effects of the modeling results. This may affect the DRF results hence an explanation was added in the revised manuscript in lines 3-6, page 10: "Note that there are several factors that may contribute to the uncertainty in estimating the BC DRF, such as underestimation of the LRT contribution due to the current model vertical set-up, missing sources and other uncertainty in the EI data and so on. The simulated BC DRF results may be underestimated that need to be addressed in future studies".

10) P9. L19. The authors mentioned the different between BY2007 and 2030 are listed 1.2, 2.4 and 4.3 ug/m3. These results seem falling into the uncertainty range of the results. As shown in Figure 2 to 5 in the part 1, the modeling errors of BC and PM2.5 are huge (1-5 times lower than observed). Can the authors elaborate more on that? How this underestimations may influence the results on health impact analysis. In

P5 L13, "CR data was obtained from Smith who indicates that every increase in PM2.5 by 1 ug/m3 is approximately associated with an 1.006% increase in the risk." How the model uncertainty affects the risk calculation?

Response: Thank you for the important comment. We recognize this experiment still has many uncertainties, such as those related to the EI and emission projections. However, the limited observation data for the performance evaluation is another issue especially for PM2.5. Therefore, all the uncertainties will be translated into the uncertainty of the health effect results even though we used in the calculation of health risk the "$\Delta$C", i.e. not the absolute concentration. The change may reflect the impact of the intervention (emission reduction).

A discussion was added in lines 25-30, page 11: "The uncertainty of modeled PM2.5 and BC was caused by several factors such as missing sources and other uncertainty in the EI data, incorporation of LRT, grid average to point-based observation and so on. In addition, limited observation data available have prevented a more comprehensive model performance evaluation. These all would be translated into the uncertainty of the health effect results. Even though we used change in the annual ambient PM2.5 concentration ($\Delta$C) in the calculation of health risk, the impact of the intervention (emission reduction) may still be analyzed with a high uncertainty".

A recommendation for future studies to improve the results of premature mortality by generating the regional/country specific data such as Concentration Response (CR) function for PM2.5 has been added in lines 30-32, page 11: "Further, regional/country specific CR data for PM2.5 should be generated and the uncertainties related to the modeling results should be reduced to improve the impact assessment of emission reduction scenarios on premature mortality in the SEA region".

11) Table 1. Header for PM1. And PM2. Were not showing properly.

Response: Thank you for the correction. We checked Table 1 and it seems OK. Do you mean with Figure S2?. The figure was refined accordingly in the revised version.

12) Table 2. Co-benefits of emission reduction? What kind of co-benefits? I think the title should be "summary of emission reduction scenarios for the SEA domain". The value of "327 and 472 ug/m3 for hourly maximum seems to be very large. Please double check.

Response: Thank you. We revised the Table 2 caption to "Summary of emission reduction scenarios for the SEA domain".

We meant co-benefits for reduction in air pollutant concentrations (i.e. associated impact of PM concentration reduction on the premature mortality) and the reduction in the BC DRF. The hourly maximum PM10 concentration of 327 $\mu$g/m3 occurred in the Borneo Island during the intensive period of biomass open burning, which should be realistic. The maximum concentration of 472 $\mu$g/m3 also occurred in the same place where the emission from crop residue open burning was assumed to be intensified under the BAU2030.

Discussion has been now added in the revised MS lines 23-30, page 10: "The simulated maximum hourly concentrations of PM2.5, PM10, and BC under BY2007 were 189, 327 and 39 $\mu$g m-3, respectively that increased to 296, 472, and 59 $\mu$g m-3 under the BAU2030. Measures implemented under the RED2030 helped to reduce the hourly maximum concentrations of PM2.5, PM10, and BC to 146, 247, 32 $\mu$g m-3 (Table 2). The hourly maximum concentrations of PM2.5 and PM10 occurred in the Borneo Island during the intensive period biomass open burning while for BC it occurred in over the eastern part of Java Island. Sharp increase in the simulated maximum hourly concentrations under the BAU2030 (e.g. PM10 of 472 $\mu$g/m3) also occurred in the same place (Borneo Island) where the emission from crop residue open burning was assumed to be intensified".

13) Figure 3. Very strange to see areas outside of Jakartar would have the same impacts as Jackartar. As shown in Figure 2, ] high concentrations of PM2.5 and BC are found in Jakartar, not other places in the island. However, the mortality cases in

Figure 3 are all red for the island.

Response: Thank you for the suggestion. We improved the plot and color scale for the high mortality rate in the revised version of the manuscript (Figure 3).

Please also note the supplement to this comment:
https://www.atmos-chem-phys-discuss.net/acp-2017-316/acp-2017-316-AC1-supplement.pdf

[revised manuscript text omitted]

---

## Author Response (AR1)

This is the second part of the two parts paper on the BC emission and its impact on air pollution in Southeast Asia. In this manuscript, the authors examined the impact of different scenarios of BC emission on air pollution and climate. However, the meteorological elds are xed with year 2007. Therefore, air pollution and aerosol radiative forcing as well as the number of premature death are largely proportion to the levels of BC emissions. I found this paper doesn't add any new insight to the problem. So, I do not recommend this paper for publication.

**Response:**

*Thank you for your comment. We agree that the approach is standard but we also argue that our key contribution in this part was to develop the emission scenarios taking into account the country specific data (BAU2030) and relevant policies for future emission (RED2030, emission reduction) hence could capture the realistic development of the region. We did not rely on the IPCC method in development of the emission scenarios but when we compared the emission scenarios results with the IPCC we found that the RED2030 is very much aligning with the IPCC RCP2.6 hence suggesting that the current master plans in the considered countries (Thailand and Indonesia) could lead to achieving 2.0 degree target. It is detailed in the revised manuscript lines 37, page 7 and lines 1-2, page 8:* "As compared to BY2007, the regional BC and $CO_2$ emissions under BAU2030 in 2030 increased by about 1.89 and 1.41 times, respectively, and were well above the increase rate specified by the Intergovernmental Panel on Climate Change (IPCC) RCP6.0 for BC and $CO_2$ of 1.03 and 1.23 times for SEA".

And *lines 8-13, page 9:* "It is interesting to note that our results for the national BC RED2030 to BY2007 emission ratios for Indonesia (0.74) and Thailand (0.81) were close to the IPCC RCP2.6 scenario for the 2 countries of 0.82 and 0.81, respectively. For $CO_2$, the emission ratios were 0.7 and 0.84 which were slightly lower than the IPCC RCP2.6 values of 0.92 and 0.86, respectively. It therefore shows that the RED2030 is very much aligning with the IPCC RCP2.6 for the countries and therefore suggesting that the current master plans in the two countries could lead to achieving the 2.0º target".

*This study, to our best knowledge is the first that use modeling to analyze the co-benefits at the SEA regional scale. The results therefore would contribute scientific evidence to promote the co-benefit approach that is currently not incorporated in the policy in any country in the region. We have now highlighted this point in our revised MS, lines 1-4, page 3:* "To our best knowledge, this is the first study addressing co-benefits for the SEA region hence the results would contribute scientific evidences to promote the co-benefit approach that is currently not incorporated in the policy in any country in the region".

*We also agree that it would be interesting to use the future meteorology for the 2030 simulation. However, we selected to use 2007 meteorology because the WRF performance was shown acceptable in Part I (now titled as* "Integrated emission inventory and modeling to assess distribution of particulate matter mass and black carbon composition in Southeast Asia"). *It is therefore would be useful to check the impact of the emission reduction alone on the climate forcing and health effects in SEA using the WRF evaluated results. Our focus was to examine the*

*impact of emission reduction measures hence we selected to use the tested meteorology (fixed on 2007) for the SEA. Nevertheless, we incorporated the future (2030) boundary conditions from the global CTM simulation of LMDz-INCA to account for the future inflows of pollutants from outside the domain. We recognize the importance of the incorporation of the future meteorology and multiyear simulations hence added this as a recommendation in the revised MS lines 23-25, page 13: "Multiyear simulations using an on-line coupled climate – chemistry modeling system should be conducted to provide a more realistic impact resulting from emission reduction scenarios on air quality and climate in SEA".*

Speci c Comments:

1) Experiment setting is not clearly described. If I understood correctly, all meteorological variables and SST are same as year-2007 (lines 8-9 on page 3). Only emissions and chemistry boundary conditions are changed. This modeling experiment doesn't represent the future scenarios.

**Response:**

*Thank you. As noted above, we kept all meteorological variables and SST the same of 2007 to test the impacts of the emission changes in SEA. We agree that our experiment focused on the emission scenarios.*

2) Is DRF for all-sky or clear-sky?

**Response:**

*The DRF was calculated for clear-sky. The sentence was revised to add the clear-sky term in the revised MS in lines 18-19, page 6: "The expression of (1-Ac) represents the clear sky conditions, i.e. we consider only clear sky BC DRF".*

*More on the uncertainty due to the use of clear sky DRF is also discussed in response to comment #4 below.*

3) Lines 24-26 on page 3: I understand that long-term data record is not available for many cities, but 4-19 years are too short to determine trend or BAU scenario.

**Response:**

*We realized the short period would introduce bias for the Business as Usual (BAU) scenario. However, we are limited with the availability of data for the two considered countries (Indonesia and Thailand). In addition, in 1997-1998 the region has gone through the drastic economic recession hence inclusion of the data before 1999 may introduce more bias also in the projection. Therefore, even when longer data period was available we still relied on the trend obtained for the period 1999-2007 for the sectors. Exception was for the petroleum refinery and mining production in Thailand that had only a short period of historical data (4 years) hence it may contribute to the uncertainty of the BAU scenario development. Luckily, these sectors do not contribute significantly to the total emissions.*

*This explanation is now added in the revised MS lines 2-8 page 4: "In principle, it is desirable to have long historical data periods for the emission projection (for Thailand and Indonesia). However, in 1997-1998 the region went through a drastic economic recession hence inclusion of the data before 1999 may induce a large bias. Therefore, even when a longer data period available for a sector we still relied on the trend obtained for the period of 1999-2007. Note that only a short period of historical data (4 years) available for the petroleum refinery and mining production sectors in Thailand hence it may contribute more uncertainty to the BAU scenario emission results. Nevertheless, these sectors did not contribute significantly to the total emissions".*

*Footnote has been also added in the Table S2: "Note: only the 1999-2007 data period was used for the regression analysis. The data available before that was not incorporated because of the drastic economic recession in SEA during 1997-1998".*

4) Line 34 on page 10, "bring in substantial bene t to human health and climate": Here, by the bene t to climate, the author seem to indicate the reduction of BC DRF due to reduction of BC emission. First of all, could you clarify whether DRF is for all sky or clear sky? Also, the reduction does not necessarily mean bene cial. How to de ne bene cial for this case, climate? Also, it is not climate, but climate forcing.

**Response:**

*The DRF was for clear sky only as also in the response to the previous comment #2. We agree that the calculation for clear sky only may add uncertainty to the results. We added this in the revised MS lines 18-22, page 12, and also changes to "climate forcing reduction, as per your suggestion.*

*"Our study thus demonstrated that the measures implemented to reduce BC (and PM) under RED2030 may bring in substantial benefits to avoid premature mortality and reduction in the BC DRF. Future studies should focus on the improvement of emission inventory data, modeling set-up, as well as the inclusion of the cloud microphysics in the radiative forcing calculation. The climate feedback should be expressed as the temperature changes in the future studies".*

5) Figure S1: Dotted line (=ratio of 1.0) does not match with color bars. For example, dotted line in the maritime continent is not same as that along the east coast of India.

**Response:**

*Thank you for your comment. There was a plotting problem. We now removed the dotted contour lines for BC and OC in Figure S1 as it is already explained in the color scale.*

Anonymous Referee #2

The study investigates the co-bene ts of black carbon and PM2.5 emission reductions from climate and health related perspectives. Part 1 summaries the development of South East Asia (SEA) emission inventory, and validation of air quality model, while part 2 summaries the climate and health bene ts from the reductions of BC and PM2.5 in 2030 for SEA. Overall, the paper is written within the scope of ACP and in a reasonable quality. Major changes, rerun and more in-depth explanation should be done before accepting for ACP.

**Response:**

*Thank you for your comments and suggestions.*

General comments:

1) The experimental plan was well established. However, the model implementation (particularly in the air quality part) was not well delivered. It is necessary to redo the simulation to resolve that. For example (described in part 1), CHIMERE only uses 8 vertical layers (up to 5.5 km). How the authors can consider long-range transport of air pollutants (e.g., PM2.5 and BC), which often happens at the free troposphere (from 2 km to 16 km – tropic). How the authors come up AOD only with 5.5 km column? Although I understand that the vertical pro le of aerosol concentration goes down as increase of height. However, AOD is measured based on the entire vertical column. The in uence of missing mid and top- tropospheric (i.e., about 10 km) concentration could be huge, even with low aerosol concentrations. This could introduce a huge bias to the results. Moreover, major biomass burning in uence occurs at the free troposphere, as plume-rise normally bought biomass burning BC and PM2.5 up to 6-10km. Without upper layer information, I am not sure how the authors can consider that. I agree with the authors that for the anthropogenic emissions, the in uence of upper layer may not have much impact to the surface. That is only limited to the case with "no long-range transport", "no biomass burning", and "no tropospheric downwash.

Response:

*Thank you for your input and we agree that, with the current vertical set-up, the long-range transport (LRT) that takes place in the free tropospheric layer may not be captured hence causing bias in the total column AOD and DRF results when LRT is substantial. However, because of the Planetary Boundary Layer (PBL) in our domain showed the maximum hourly value of 3,900 m, i.e. below the upper level of the domain (5,500 km), hence the mixing down effects of the LRT pollution from above PBL may be partly included. This has been explained in our accompanying paper (Part 1, now titled as "*Integrated emission inventory and modeling to assess distribution of particulate matter mass and black carbon composition in Southeast Asia*").*

*We used the previous version of the model (CHIMERE 2008c) with a simplified photolysis rate calculation (Madronich et al., 1998) that assumed the model domain is below the cloud, hence putting a constraint on model top to be maximum at 500 hPa. In our accompanying paper (Part I), we recommended for the future studies to use the most recent version of the model with updated*

*radiative transfer model hence can compute photolysis reaction rates for extending the model domain vertically beyond 500 hPa. Regarding the biomass open burning, it was found that the aerosol extinction coefficient (in the transboundary haze affected sites in SEA observed by the level 2 NASA Cloud Aerosol Lidar with Orthogonal Polarization (CALIOP)) was not significant above the height of 5 km (Campbell et al., 2013). Thus, the uncertainty caused by the vertical model set-up is expected but the effect may be small.*

*We have added a discussion in the revised MS in lines 22-28, page 6 "Note that, in the current vertical model set-up the top of the model domain was 500 hPa hence the simulation results may not be able to capture the long-range transport (LRT) that took place above the domain top in the free tropospheric layer and subsequently underestimate BC AOD and BC DRF. However, above the domain top (>5 km) the biomass open burning emission of importance in the region may not have significant effects on the aerosol extinction coefficient as shown by the Cloud-Aerosol Lidar with Orthogonal Polarization (CALIOP) observations (Campbell et al., 2013). Therefore, the uncertainty caused by the vertical model set-up on the simulation results may not be significant considering the biomass open burning emissions in particular".*

2) Be honest, model performance is very bad for 30 km x 30 km resolution. For example, in part 1 Table 3, the RMSE for some of the sites is more than 10 degree Celsius. The highest could reach 20.5 degree Celsius. I am not sure whether I can trust these results. Please double check the statistic results.

Response:

*Thank you for your correction. We checked the results and the situation was for the 2 stations in the Philippines that have large missing data and these were taken as "zero" in the statistical measure calculation (e.g. RMSE) template used. In the revised version of Part I we recalculated the statistical measures using only the hours with observation data and results (Table 3, Part I) showed better performance. We also referred to Part I in the discussion.*

3) In order to understand the part 2, more description of the regridding process, and what spatial interpolation methods (e.g., spatial surrogate) used in the study for generating the model emissions are important, as the countries in marine time continents are widely spread. How authors use those regional emissions from 50 km to 1 degree emission results (e.g., EDGAR or CGRED) to derive 30 km x 30 km resolution for the simulation.

Response:

*The emissions from EDGAR and CGRER are available at 0.5° x 0.5° (50 x 50 km$^2$) resolution. The EI we prepared for the three countries had a resolution of 30 x 30 km$^2$. We used the spatial analysis of "grid conversion" in Geographical Information System (GIS) to overlay the emission with the land mask map (generated from WRF, resolution of 30 x 30 km$^2$) to specially regrid the border between land and sea.*

*We added information in the revised MS lines 13-17, page 4: "The BY2007 emission data for those countries was taken from the international databases that were provided at a different resolution (i.e. 0.5º x 0.5º). Therefore, the spatial analysis of "grid conversion" in Geographical Information System (GIS) was used to convert the emission data in to the same grid resolution 30 x 30 km² for the whole domain. The land mask map that was generated from WRF model with a resolution of 30 x 30 km² was used to regrid the emissions in the border between land and sea (assigning zero emissions to the sea area)".*

4) The information from LMDZ/INCA was insuf cient. I can't nd enough information to understand the process. Any species mapping table, or cross-referencing table is available.

Speci c comments

1) P1. L 21, more information on LMDZ/INCA.

Response:

*Information on the LMDz/INCA model was added in the revised MS in lines 14-18, page 3: "However, to account for the changes in the boundary conditions in the simulations of future emission scenarios, we used the chemistry boundary conditions obtained from the global chemistry–aerosol– climate model that couple online the LMDz (Laboratoire de Météorologie Dynamique, version 4) general circulation model (Hourdin et al., 2006) and the INCA (INteraction with Chemistry and Aerosols, version 3) model (Hauglustaine et al., 2014) using the global RCP8.5 emissions (Hauglustaine, 2013)".*

*And lines 21-22, page 3: "The horizontal resolution is 1.9º in latitude and 3.75º in longitude of LMDZ/INCA output".*

*However, since we obtained the results from the simulation conducted by Dr. Didier Hauglustaine (CNRS-LSCE) who was acknowledged in the MS, we did not present the results of LMDz/INCA in detail but added references for the publications that describe the model detail configurations.*

P2. L21-23, please rewrite the sentence, I don't fully understand.

Response:

*The sentence has been re-written as follows (lines 37-38, page 2 and line 1. Page 3 ):*

*"The results of this study would provide information to policy makers on the efficacy of different emission reduction measures and associated co-benefits for improving air quality, reducing health effects, and mitigating climate forcing in SEA".*

2) P3. L7, why only 1 year of WRF? Is 2007 a typical average year? Is it a wet or dry year? Normally, I will do three years simulation for any climate related study since it has to take into consideration of the interannual variability. Of course, this study relates more on

air quality. I can understand using 1 year data. However, the authors may need to demonstrate 2007 is an average typical year, in terms of temperature, and precipitation. Sometimes, ENSO effect may have a huge impact, and may bias the results in the marine-time continents.

Response:

*We focused on the air pollutants that have short lifetime in the atmosphere, i.e. BC that is a short-lived climate forcing pollutant (SLCPs), hence considered only 1 year simulation. However we also recognize that longer term with multiyear simulation would provide more representative results. Therefore we added a recommendation in this MS (also to reflect the reviewer 1 comment) in lines 23-25, page 13:* "Multiyear simulations using an on-line coupled climate – chemistry modeling system should be conducted to provide a more realistic impact resulting from emission reduction scenarios on air quality and climate in SEA".

*The year of 2007 was selected because it was not affected by the strong El Niño and La Niña. The Oceanic Niño Index (ONI) analyses found that the year 2007 was categorized as weak La Niña (http://www.cpc.noaa.gov/products/analysis_monitoring/ensostuff/ensoyears.shtml). The information has been added in the revised manuscript in lines 11-12, page 3:* "The base year of 2007 was selected because it was not affected by the strong El Niño and La Niña events (http://www.cpc.noaa.gov/products/analysis_monitoring/ensostuff/ensoyears.shtml)".

3) P3. L13, global LMDZ/INCAS has 19 layers, how the authors interpolate the results into 8 layers CHIMERE results. Is the 2030 case using 2030 boundary conditions from LMDZ/INCAS. The part 1 paper didn't mention anything related 2030 scenarios for LMDZ model. More information should be provided.

Response:

*The script of prep_bound.f90 (one routine in CHIMERE) read the monthly concentrations from the global CTM of LMDz/INCA and get the information of the lateral (lat_con) and top boundary (top_con) concentrations. The vertical profile of the concentration was interpolated to the similar layer (8 layers) set in the CHIMERE simulation. The results for 7 layers in the stratosphere (included in the LMDz/INCA simulation results) were not included in the boundary condition processing. More information was added in the revised manuscript for the 2030 simulation of LMDz in lines 22-24, page 3:* "Boundary conditions from the LMDz/INCA were processed using the available routine in CHIMERE to read the monthly concentrations and get the information of the lateral (lat_con) and top boundary (top_con) concentrations".

*In Part 1 paper, we focused on the model performance evaluation for the base year of 2007 hence not mentioned about scenarios which are the focus of this paper actually. Now we added a sentence in the conclusions introducing this paper in lines 27-29, page 12:* "This study is a continuation of our previous paper (Permadi et al., 2017a focusing on the model performance evaluation for the BY2007) to present the development of two emission scenarios for SEA in 2030, BAU2030 and RED2030, and assess the associated impacts on the premature mortality and climate forcing in the region".

5) Moreover, for the Figure S1, the description should be 2030/2007, not 2030/2006. Also, which case is it for 2030? BAU, RED? Very unclear. Annual concentration or monthly average (Jan, Aug)?

Response:

*Thank you for the correction. It was a typo, we corrected the table footnote to be* 2030/2007 *and the figure caption of "Annual average…" has been corrected to "*Monthly average*…." In Figure S1. The figure was modified and now only presents the ratios between simulated BC (and OC) for 2007 and 2030 by the LMDz/INCA (RCP8.5) that were used in our study for estimating the boundary conditions. The LMDz/INCA global simulation for RCP8.5 (including SEA) was used for the extraction of the boundary conditions. We used the ratio of 2030/2007 produced by LMDz/INCA for each species to scale up the boundary conditions from 2007 to 2030.*

*An explanation has been added in the revised MS lines 24-25, page 3: "*We used the ratio of simulated levels between 2030 and 2007 for each species to estimate the boundary conditions in 2030 and modified the model inputs*".*

6) P3. L31, Yes, same EFs for 2007 were used for 2030 may contribute a certain uncertainty to the projected emission. Will cause increase or decrease? Can the authors provide more explanation?

Response:

*Thank you for the input. We agree that using the same emission factors contributes to the uncertainty of the projected emissions. In general, we expect the emission factors to be reduced when new technologies are available in 2030. However, if the age limit is not enforced for vehicles then aged engine would have higher EF in the future. An explanation was added in the revised manuscript lines 10-12, page 4: "*In general, we should expect the EFs to be reduced in the future with the progressive technology intrusion. Nevertheless, if the vehicle age limit is not strictly enforced for then more aged engines would have higher EFs in the future*".*

7)P3. L31, Is the projection align well with IPCC projection for 2030 for those local projections?

Response:

*We did not follow the IPCC Representative Concentration Pathway (RCP) scenarios in developing the scenarios for Indonesia and Thailand hence we do not expect 100% aligning. We found that under the IPCC RCP8.5 scenario for SEA domain, found a reduction in BC emission by 3.12% as compared with 2007 while under IPCC RCP6.0 an increase by 2.7% (1.027 times) was shown.*

*As compared to 2007, our BAU2030 increased the regional BC and $CO_2$ emissions in 2030 and were well above the increase rate under IPCC RCP6.0 for SEA domain. We added this comparison in the revised MS, line 37, page 7 and lines 1-2, page 8: "*As compared to BY2007, the regional BC and $CO_2$ emissions under BAU2030 in 2030 increased by about 1.89 and 1.41 times,

respectively, and were well above the increase rate specified by the Intergovernmental Panel on Climate Change (IPCC) RCP6.0 for BC and $CO_2$ of 1.03 and 1.23 times for SEA".

*Our national BC and $CO_2$ emission ratios (RED2030/BY2007) were close to the RCP2.6 scenario and discussion has been added in the revised MS in lines 8-13, page 9: "It is interesting to note that our results for the national BC RED2030 to BY2007 emission ratios for Indonesia (0.74) and Thailand (0.81) were close to the IPCC RCP2.6 scenario for the 2 countries of 0.82 and 0.81, respectively. For $CO_2$, the emission ratios were 0.7 and 0.84 which were slightly lower than the IPCC RCP2.6 values of 0.92 and 0.86, respectively. It therefore shows that the RED2030 is very much aligning with the IPCC RCP2.6 for the countries and therefore suggesting that the current master plans in the two countries could lead to achieving the 2.0º target".*

8) P6. L34, recent years, Thailand has started restrict local burning. How this may affect the projection?

Response:

*We included the scenario of the policy of Thailand government to ban the open burning in the RED2030 by reviewing the National Masterplan to Control Open Burning (MS, lines 36-38, page 4). We took into account the measure to limit the forest area burned not over 48,000 ha/yr which has been included in the manuscript (Table S1).*

9) P8. L25, as mentioned from the general comments, only 8 layers (up to 5.5km) may not cover the entire vertical pro le. What may be the impact on this? Also, from the part 1 of the paper, the BC was well underestimated (Figure 8 in part 1 using AERONET data). How this underestimations of BC and PM2.5 would affect the results on the analysis of direct radiative forcing?

Response:

*We realize the underestimation of PM and BC and that may be caused by several factors: 1) model vertical set-up, 2) uncertainty (e.g. missing sources) in EI, and 3) meteorological parameters, as well as gridded averaging effects of the modeling results. This may affect the DRF results hence an explanation was added in the revised manuscript in lines 32-35, page 11: "Note that there are several factors contributing to the uncertainty in estimating the BC DRF, such as underestimation of the LRT contribution due to the current model vertical set-up, missing sources and other uncertainty in the EI data. Further, BC DRF was calculated for clear sky conditions therefore effect of BC on the cloud microphysics (Chýlek et al., 1996) was not incorporated".*

10) P9. L19. The authors mentioned the different between BY2007 and 2030 are listed 1.2, 2.4 and 4.3 ug/m3. These results seem falling into the uncertainty range of the results. As shown in Figure 2 to 5 in the part 1, the modeling errors of BC and PM2.5 are huge (1-5 times lower than observed). Can the authors elaborate more on that? How this underestimations may in uence the results on health impact analysis. In P5 L13, "CR data was obtained from Smith who indicates that every increase in PM2.5 by 1 ug/m3 is approximately associated with an 1.006% increase in the risk." How the model uncertainty affects the risk calculation?

Response:

*Thank you for the important comment. We recognize this experiment still has many uncertainties, such as those related to the EI and emission projections. However, the limited observation data for the performance evaluation is another issue especially for PM$_{2.5}$. Therefore, all the uncertainties will be translated into the uncertainty of the health effect results even though we used in the calculation of health risk the "ΔC", i.e. not the absolute concentration. The change may reflect the impact of the intervention (emission reduction).*

*A discussion was added in lines 33-37, page 10: "The uncertainty of modeled PM$_{2.5}$ and BC was caused by several factors such as missing sources and other uncertainty in the EI data, incorporation of LRT, grid average to point-based observation and so on. In addition, the limited observation data available have prevented from a more comprehensive model performance evaluation. These all would be translated into the uncertainty of the health effect and radiative effect results. Even though we used the change in the annual ambient PM$_{2.5}$ concentration (ΔC) in the calculation of the health risk, the resulting impact of the intervention (emission reduction) may still contain a high uncertainty".*

*A recommendation for future studies to improve the results of premature mortality by generating the regional/country specific data such as Concentration Response (CR) function for PM$_{2.5}$ has been added in lines 1-3, page 11: "Further, the regional and/or country specific CR data for PM$_{2.5}$ should be developed to improve the impact assessment of emission reduction scenarios on the premature mortality".*

11) Table 1. Header for PM1. And PM2. Were not showing properly.

Response:

*Thank you for the correction. We checked Table 1 and it seems OK. Do you mean with Figure S2?. The figure was refined accordingly in the revised version.*

12) Table 2. Co-bene ts of emission reduction? What kind of co-bene ts? I think the title should be "summary of emission reduction scenarios for the SEA domain". The value of "327 and 472 ug/m3 for hourly maximum seems to be very large. Please double check.

Response:

*Thank you. We revised the Table 2 caption to "Summary of emission reduction scenarios for the SEA domain".*

*We meant co-benefits for reduction in air pollutant concentrations (i.e. associated impact of PM concentration reduction on the premature mortality) and the reduction in the BC DRF. The hourly maximum PM$_{10}$ concentration of 327 μg/m$^3$ occurred in the Borneo Island during the intensive period of biomass open burning, which should be realistic. The maximum concentration of 472 μg/m$^3$ also occurred in the same place where the emission from crop residue open burning was assumed to be intensified under the BAU2030.*

*Discussion has been now added in the revised MS lines 32-37, page 9 and lines 1-2, page 10: "The simulated maximum hourly concentrations of $PM_{2.5}$, $PM_{10}$, and BC under BY2007 were 189, 327 and 39 µg m$^{-3}$, respectively, that increased to 296, 472, and 59 µg m$^{-3}$, respectively, under the BAU2030. Measures implemented under RED2030 would reduce the hourly maximum concentrations of $PM_{2.5}$, $PM_{10}$, and BC to 146, 247, 32 µg m$^{-3}$ (Table 2). The hourly maximum concentrations of $PM_{2.5}$ and $PM_{10}$ occurred in the Borneo Island during the intensive period of biomass open burning while that for BC occurred over the eastern part of Java Island. A sharp increase in the maximum hourly concentrations under BAU2030 (e.g. 1h $PM_{10}$ reached 472 µg/m$^{3}$) also occurred in the Borneo Island where the emissions from crop residue open burning were assumed to be intensified".*

13) Figure 3. Very strange to see areas outside of Jakartar would have the same impacts as Jackartar. As shown in Figure 2, ] high concentrations of PM2.5 and BC are found in Jakartar, not other places in the island. However, the mortality cases in Figure 3 are all red for the island.

Response:

*Thank you for the suggestion. We improved the plot and color scale for the high mortality rate in the revised version of the manuscript (Figure 3).*

---

## Author Response (AR2)

Reviewer 1 #

I found the authors address the various limitations of this study more clearly in the revised/resubmitted manuscript. One thing I still can not agree with is the "co-benefit" on climate forcing. By "co-benefit", the authors refer the reduction in BC DRF in clear sky. First of all, the modeling set-up doesn't seem to allow interaction between aerosols (CHIMERE) and meteorology (WRF). Second, the background meteorology does not represent the condition in year 2030. Third, the BC DRF is calculated for clear sky. So the reduction in BC DRF just due to the reduction in BC loading (no change in climate & no aerosol-weather interaction). I suggest to change "co-benefits" in the title to "effects" or "impacts".

Response:

*Thank you very much for your valid insights and suggestion. We fully agree to change the word "co-benefit" to "impacts" in the revised title and also in the text. The title was revised to "Assessment of emission scenarios for 2030 and* **impacts** *of black carbon emission reduction measures on air quality and* **radiative** *forcing in Southeast Asia".*

*The word of co-benefit was revised throughout the manuscript:*

- *Line 10-13, page 1, "Our accompanying paper (Permadi et al., 2017a) focuses on the preparation of emission input data and evaluation of WRF/CHIMERE performance in 2007, this paper follows with detailing the* impacts assessment *of the future (2030) black carbon (BC) emission reduction measures for Southeast Asia (SEA) countries* on air quality, health and BC direct radiative forcing (DRF)".*
- *Line 21-23, page 1, "Under RED2030, the health* benefits *were analyzed in term of the avoided number of premature deaths associated with ambient PM$_{2.5}$ reduction* along *with BC DRF reduction".*
- *Line 27-28, page 1, "Substantial* impacts *on human health and BC* DRF *reduction in SEA could be resulted from the emission measures incorporated in RED2030".*
- *Line 28-30, page 1, "Future works should consider other* impacts *such as for the agricultural crop production as well as the cost benefit analysis of the measures implementation to provide relevant information for policy making".*
- *Line 34-36, page 2, "The changes in the BC direct radiative forcing (DRF) and in the number of avoided premature deaths between BY2007 and RED2030 were compared to those between BY2007 and the business as usual scenario (BAU2030) to highlight potential* impacts".*
- *Line 36-38, page 2, "The results of this study would provide information to policy makers on the efficacy of different emission reduction measures and associated* benefits *for improving air quality, reducing health effects, and mitigating* BC DRF *in SEA".*
- *Line 38, page 2 – line 3, page 3, "To our best knowledge, this is the first study addressing air quality and* BC DRF impacts *for the SEA region hence the results would contribute scientific evidences to promote the* co-control *approach that is currently not incorporated in the policy in any country in the region".*

- *Line 8, page 5, "**2.3 Assessment of impacts on air quality and BC direct radiative forcing**"*
- *Line 9-10, page 5, "The potential impacts of the emission reduction scenarios on improvement of air quality (hence health benefits) and mitigation of climate forcing were assessed and quantified".*
- *Line 14, page 9, "**3.2 Impacts assessment of emission reduction measures in 2030**"*
- *Line 15-16, page 9, "Our impacts assessment of emission reduction measures in 2030 covered the health impact in term of the avoided number of premature deaths associated with the reduced $PM_{2.5}$ pollution and the reduction in BC DRF".*
- *Line 37, page 10, "These all would be translated into the uncertainty of the health and BC DRF effect results".*
- *Line 18-19, page 12, "Our study thus demonstrated that the measures implemented to reduce BC (and PM) under RED2030 may bring in substantial benefits in avoiding the premature mortality and reduction in the BC DRF".*
- *Line 22-24, page 12, "The impacts on crop production and materials should also be considered and the monetary values of the benefits should be presented to better inform policy makers and to promote mitigation measures for the SLCPs".*
- *Line 27-29, page 12, "This study is a continuation of our previous paper (Permadi et al., 2017a, focusing on the model performance evaluation for the BY2007) and presents the development of two emission scenarios for SEA in 2030, (BAU2030 and RED2030) to assess the associated impacts on the premature mortality and BC DRF in the region".*
- *Line 8-10, page 13, "WRF/CHIMERE/AODEM modelling system simulation results provided the PM ambient concentrations (i.e. $PM_{2.5}$, $PM_{10}$, and BC), AOD and BC DRF under different scenarios which showed substantial benefits of the emission reduction under RED2030 in improving regional air quality and BC DRF reduction".*
- *Line 20-21, page 13, "Other pollutants (beside PM and BC) should be included in the assessment of health impacts".*

I recommend this manuscript for publication only if the authors de-emphasize the co-benefit on climate forcing.

Response:

*Thank you for your recommendation and we agree to de-emphasize the co-benefit on climate forcing reflected in the revised MS.*